# PolSAR Models with Multimodal Intensities

**Jodavid A. Ferreira** [1,†] , **Abraão D. C. Nascimento** [2] and **Alejandro C. Frery** [3,*,†]

1   Departamento de Estatística, Universidade Federal da Paraíba, João Pessoa 58051-900, PB, Brazil
2   Departamento de Estatística, Universidade Federal de Pernambuco, Recife 50670-901, PE, Brazil
3   School of Mathematics and Statistics, Victoria University of Wellington, Wellington 6140, New Zealand
*   Correspondence: alejandro.frery@vuw.ac.nz
†   These authors contributed equally to this work.

**Abstract:** Polarimetric synthetic aperture radar (PolSAR) systems are an important remote sensing tool. Such systems can provide high spacial resolution images, but they are contaminated by an interference pattern called multidimensional speckle. This fact requires that PolSAR images receive specialised treatment; particularly, tailored models which are close to PolSAR physical formation are sought. In this paper, we propose two new matrix models which arise from applying the stochastic summation approach to PolSAR, called compound truncated Poisson complex Wishart (CTPCW) and compound geometric complex Wishart (CGCW) distributions. These models offer the unique ability to express multimodal data. Some of their mathematical properties are derived and discussed—characteristic function and Mellin-kind log-cumulants (MLCs). Moreover, maximum likelihood (ML) estimation procedures via expectation maximisation algorithm for CTPCW and CGCW parameters are furnished as well as MLC-based goodness-of-fit graphical tools. Monte Carlo experiment results indicate ML estimates perform at what is asymptotically expected (small bias and mean square error) even for small sample sizes. Finally, our proposals are employed to describe actual PolSAR images, presenting evidence that they can outperform other well-known distributions, such as $\mathcal{W}_m^{\mathbb{C}}$, $\mathcal{G}_m^0$, and $\mathcal{K}_m$.

**Keywords:** matrix model; stochastic summation; Mellin; EM

## 1. Introduction

Polarimetric synthetic aperture radar (PolSAR) systems are an efficient remote sensing tool. Electing two among its many benefits, the use of these systems is justified by their capability to operate under various weather conditions and of producing relevant amounts of information about the target. However, PolSAR images are contaminated by a multidimensional interference called speckle, which manifests as a multiplicative behavior and a significant degree of interference on the generated data. These issues preclude both the direct use of classical image processing techniques (which often assume additivity and Gaussianity) and the visual interpretation of textures and targets. As a consequence, defining adequate PolSAR models is a crucial step for several applications in polarimetry, such as classification [1,2], segmentation [3], speckle filtering [4,5], and boundary and change detection [6,7].

This paper is focused on distributions for multilook PolSAR data, which belong to the set of positive definite hermitian matrices, say $\Omega_+$, and are sometimes referred to as sample covariance matrices (SCM) [8]. There are several multilook PolSAR models in the literature. They are often derived either from the multiplicative modeling (MM) or from the inverse Fourier transform of its probability density function.

The goal of this paper is three-fold. Firstly, we aim to derive analytically two new three-parameter probabilistic models referred to as compound truncated Poisson complex Wishart (CTPCW) and compound geometric complex Wishart (CGCW) distributions, and some of their marginal intensity laws. These marginal probability density functions are

able to express multimodal data; this unique feature had only been previously obtained by the use of mixture models. The relation of CTPCW and CGCW models to the multilook PolSAR imagery physical formation is examined as well. Then, two of their properties are derived and discussed: their characteristic functions (cf) and their Mellin-kind log-cumulants (MLCs). Secondly, we provide maximum likelihood estimators (MLEs) via the Expectation Maximisation (EM) algorithm for the CTPCW and CGCW parameters. Further, we furnish CTPCW and CGCW goodness-of-fit (GoF) graphical tools. A Monte Carlo study was conducted to quantify the performance of MLEs in terms of some figures of merit. Third, two applications to actual PolSAR data were performed. Our proposals are compared with three well-known PolSAR distributions: $s\mathcal{W}_m^{\mathbb{C}}$, $\mathcal{G}_m^0$, and $\mathcal{K}_m$. Results illustrate the importance of introduced models when analysing PolSAR imagery.

This paper is organised as follows. In Section 2, we review the literature about modeling PolSAR data, and introduce two new distributions and some of their properties. Estimation methods and GoF tools for both distributions are provided in Section 3. In Section 4, the numerical results are displayed. The main conclusions are drawn in Section 5.

## 2. PolSAR Models and Some of Their Properties

### 2.1. Literature Models and Physical Insight of Our Proposal

Taking the MM as a generator mechanism of PolSAR distributions, each returned matrix associated to an image entry is the product of two independent random variables, which describe terrain and speckle influences. For instance, assuming the $m$-dimensional scaled complex Wishart model (say $Y \sim s\mathcal{W}_m^{\mathbb{C}}$) discussed by Nascimento et al. [9] for describing the multilook multidimensional speckle noise for $m$ polarisation channels with density $f_Y$ and $X \in \mathbb{R}_+$ as a random variable for the terrain with density $f_X$, the return models $Z = YX$ holds from the general expression [10]

$$f_Z(z) = \int_0^\infty x^{-m^2} \underbrace{f_Y(z \mid x)}_{\text{Speckle}} \underbrace{f_X(x)}_{\text{Backscatter}} \, \mathrm{d}x,$$

where $z \in \Omega_+ := \{ Z \in \mathbb{C}^m \times \mathbb{C}^m : Z = Z^* \}$ is an outcome of $Z$ and $(\cdot)^*$ is the transpose conjugate operator. The subsequent framework mentions the SCM models, when $X$ follows $\mathcal{G}^0$, $\mathcal{N}^{-1}$ (inverse normal), $\Gamma^{-1}$ (reciprocal Gamma), $\Gamma$ (Gamma), beta, beta$^{-1}$ (reciprocal Beta), and $\delta$ (Dirac) distributions.

Further, we used the Laguerre expansion to extend the $\Gamma$ model for the backscatter, resulting in the generalised form of $\mathcal{K}_m$ [8]. In practice, the $s\mathcal{W}_m^{\mathbb{C}}$, $\mathcal{K}_m$, and $\mathcal{G}_m^0$ laws are commonly used for understanding ocean, forest, and urban areas, respectively. Figure 1 shows the connections among the most used PolSAR distributions. The survey by Deng et al. [11] discusses these and other SCM models. Further, Yue et al. [12] have recently presented a survey about how different assumptions for both scatter number and scattered field generate all known SAR return models and new descriptors for correlated SAR textures.

Our paper addresses two new SCM three-parameter distributions based on an extension of Ref. [12] to PolSAR returns, whose marginal laws of their main diagonals are able to describe both uni- and multi-modal events.

It is worth noting that there is a well-defined tradition of using mixtures in the processing of SAR imagery. Mentioning that the SAR return has a multiplicative nature and the gamma distribution a physical-based law for describing the speckle noise, Nicolas and Tupin [13] addressed the problem of fitting finite mixtures of gamma by means of log-cumulant theory. Krylov et al. [14] proposed a supervised classifier by combining finite mixture modelling with copulas. Solarna et al. [15] provided unsupervised change detector for multimodality SAR data assuming, as one of the pre-assumptions, the use of mixtures. Figure 1 illustrates how our proposal relates to other distributions, in particular to the $s\mathcal{W}_m^{\mathbb{C}}$ law. Table 1 summarises these distributions and provides references.

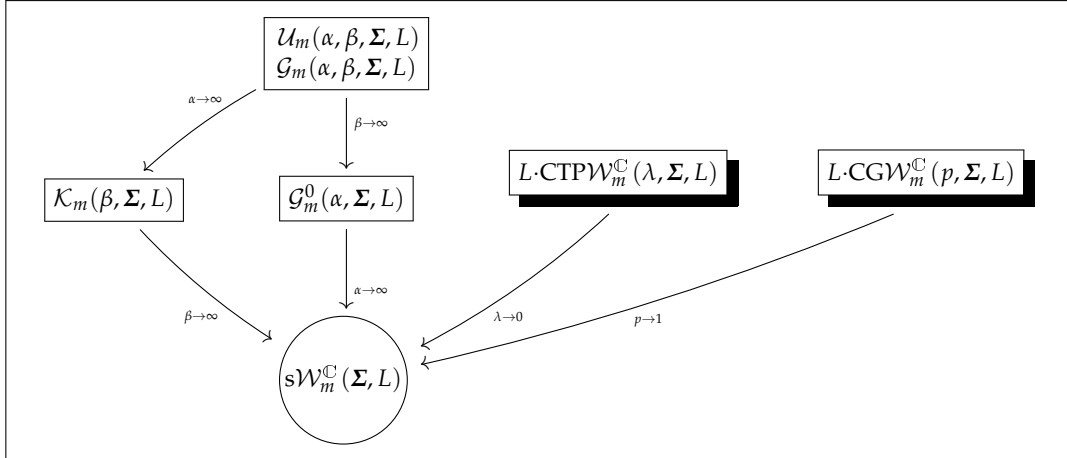

**Figure 1.** Diagram of limit relationships among PolSAR distributions. Here, the parameters $\alpha$ and $\beta$ represent shape, $\boldsymbol{\Sigma}$ denotes a kind of location, and $L$ is the equivalent number of looks (ENL).

**Table 1.** Summary of SCM distributions.

| SCM ($Z$) | Terrain ($X$) $\times$ Speckle ($Y$) | Reference |
|:---:|:---:|:---:|
| | Four-parameter models | |
| $U_m$ | $\mathcal{G}^0 \times s\mathcal{W}_m^{\mathbb{C}}$ | Bombrun and Beaulieu [16] |
| $\mathcal{G}_m$ | $\mathcal{N}^{-1} \times s\mathcal{W}_m^{\mathbb{C}}$ | Freitas et al. [17] |
| | Three-parameter models | |
| $\mathcal{G}_m^0$ | $\Gamma^{-1} \times s\mathcal{W}_m^{\mathbb{C}}$ | Freitas et al. [17] |
| $\mathcal{K}_m$ | $\Gamma \times s\mathcal{W}_m^{\mathbb{C}}$ | Lee et al. [18] |
| $W_m$ | beta $\times s\mathcal{W}_m^{\mathbb{C}}$ | Deng et al. [11] |
| $M_m$ | beta$^{-1} \times s\mathcal{W}_m^{\mathbb{C}}$ | Deng et al. [11] |
| | Two-parameter (baseline) models | |
| $s\mathcal{W}_m^{\mathbb{C}}$ | $\delta \times s\mathcal{W}_m^{\mathbb{C}}$ | Anfinsen et al. [19] |

Consider the physical explanation behind the MM approach [12]. Initially, taking only one polarisation channel, it is known that if the number of scatterers, say $N$, in one resolution cell is large enough and approximately constant, then the returned electromagnetic signal,

$$F = \sum_{k=1}^{N} F_k, \tag{1}$$

follows the complex Gaussian law [20], where $F_k$ is the complex-valued quantity representing the individual scatter. As a consequence, the amplitude, i.e., norm $\|F\| = \sqrt{F^2}$, and intensity $I = \|F\|^2$ of this signal are Rayleigh and exponential distributed, respectively.

If the number of elementary backscatterers $N$ changes among resolution cells, it must be described as a random variable. With this, the return is no longer Gaussian. According to Delignon and Pieczynski [20], if:

(i)     the random number of scatters $N$ in each elementary cell follows a Poisson distribution;
(ii)    its expected value $\mathbb{E}(N) = \lambda_0$ is itself a random variable $\Lambda$ with density $f(\lambda_0)$; and
(iii)   the density of the intensity $I$ is $g(x)$, then, adapting the original result from single-look to the $L$-look case;
(iv)    for $\lambda_0$ large enough, the conditional model $I \mid \Lambda = \lambda_0$ follows a Gamma distribution with shape $L$ and scale $(\lambda_0 \sigma^2)^{-1} L$ such that $\sigma^2$ is the common variance of the amplitude of the individual scatters; and, therefore,

(v)   the density of the unconditional intensity $I$ is

$$
\begin{aligned}
g(x) &= \int_0^\infty f_{[I|\Lambda=\lambda_0]}(x)f(\lambda_0)\mathrm{d}\lambda_0 \\
&= \int_0^\infty \frac{[L/(\lambda_0\sigma^2)]^L x^{L-1}}{\Gamma(L)} \exp\left\{-\frac{Lx}{\lambda_0\sigma^2}\right\}f(\lambda_0)\mathrm{d}\lambda_0, \\
&= \int_0^\infty \frac{1}{\lambda_0} \underbrace{f_{\Gamma(L,L/\sigma^2)}(x/\lambda_0)}_{\text{unidimensional speckle}} \underbrace{f_X(\lambda_0)}_{\text{backscatter}} \mathrm{d}\lambda_0,
\end{aligned}
$$

which is a model for each element in the main diagonal of SCM distributions.

In practice, hypothesis (iii) may not be realistic: high-resolution SAR sensors acquire a few backscatterers per resolution cell. Yue et al. [12,21] discussed the physical implications of assuming the number of scatterers as a random variable, and they derived a new model for correlated SAR textures.

In this paper, we adopt the summation model (1) to describe multilook PolSAR returns.

PolSAR systems register the amplitude and phase of backscattered signals of reception and transmission linear combinations, yielding four polarisation channels: $F_{\text{HH}}$, $F_{\text{HV}}$, $F_{\text{VH}}$, and $F_{\text{VV}}$ (H for horizontal and V for vertical polarisation). If the reciprocity theorem conditions [22] are satisfied, then $F_{\text{HV}} = F_{\text{VH}}$. Thus, multilook PolSAR returns have the form:

$$
\boldsymbol{F} = \sum_{i=1}^N \frac{1}{L} \sum_{\ell=1}^L \begin{bmatrix} F_{\text{HH},i}^{(\ell)} \\ F_{\text{HV},i}^{(\ell)} \\ F_{\text{VV},i}^{(\ell)} \end{bmatrix} \begin{bmatrix} F_{\text{HH},i}^{*(\ell)} & F_{\text{HV},i}^{*(\ell)} & F_{\text{VV},i}^{*(\ell)} \end{bmatrix},
$$

where $F_{\text{A},i}^{(\ell)} \in \mathbb{C}$ is the scattering in channel $\text{A} \in \{\text{H}, \text{V}\}$ at the $i$th individual scatterer and $\ell$-th look. Based on the results by Goodman [23] and Hagedorn et al. [24], we have evidence that the complex Wishart distribution can meaningfully represent the PolSAR return in homogeneous scenarios. Moreover, the truncated Poisson and geometric distributions are suggested to be suitable laws for modelling the returned signals within a resolution cell. Therefore, combining these two evidences, we propose the sum of a random number of complex Wishart models, with the number of terms following the truncated Poisson and geometric laws as two descriptors for the matrix-return $\boldsymbol{F}$. In what follows, we detail our contributions.

### 2.2. New Models

Assume that the random number of scatterers, $N \in \mathbb{Z}_+$, per individual cell follows one of two possible distributions: truncated Poisson [25,26], denoted $N \sim \text{TPo}(\lambda)$ with probability mass function (pmf) $\Pr(N = k) = \lambda^k / [k!(e^\lambda - 1)]$, and Geometric [27,28], $X \sim \text{Geo}(p)$ having pmf $\Pr(N = k) = p(1-p)^{k-1}$. In these cases, $\lambda > 0$ denotes the mean number of scatterers, while $p \in (0,1)$ is the probability of finding an individual scatterer. When $p \uparrow 1$ or $\lambda \downarrow 0$, these models represent the existence of one scatter per cell with probability 1.

The SAR literature has indicated the Poisson [20] and negative binomial [29] models for $N$ in the MM context, but we use the TPo and Geo laws for their analytic tractability.

First, set $N \sim \text{TPo}(\lambda)$ and $\boldsymbol{Z}_i \sim \mathcal{W}_m^{\mathbb{C}}(\boldsymbol{\Sigma}, L)$ for $i = 1, \ldots, N$ with probability density function (pdf),

$$
f(\boldsymbol{z}_i) = \frac{|\boldsymbol{z}_i|^{L-m}}{|\boldsymbol{\Sigma}|^L \Gamma_m(L)} \exp\left\{ -\operatorname{tr}\left(\boldsymbol{\Sigma}^{-1}\boldsymbol{z}_i\right) \right\},
$$

where $\Gamma_m(L) = \pi^{\frac{m(m-1)}{2}} \prod_{i=0}^{m-1} \Gamma(L-i)$ is the multivariate gamma function. Then, $S_k = \sum_{i=1}^{k} Z_i \sim \mathcal{W}_m^{\mathbb{C}}(\Sigma, kL)$ [30], and the coherence matrix per cell follows the CTPCW law, i.e., $S = \sum_{i=1}^{N} Z_i$, has pdf [31],

$$
\begin{aligned}
f(s) &= \sum_{k=1}^{\infty} \Pr(N=k) f_{S_k}(s) \\
&= \left( \frac{1}{e^\lambda - 1} \right) \sum_{i=1}^{\infty} \frac{\lambda^k}{k!} f_{\mathcal{W}_m^{\mathbb{C}}(\Sigma, kL)}(s) \\
&= \left[ \frac{e^{-\operatorname{tr}(\Sigma^{-1}s)}}{|s|^m (e^\lambda - 1)} \right] \sum_{i=1}^{\infty} \frac{\left( \lambda |\Sigma^{-1}s|^L \right)^k}{k! \Gamma_m(kL)},
\end{aligned}
$$

where $s = \{s_{i,j}\}$ is an outcome of $S = \{S_{i,j}\}$. This situation is denoted by $S \sim \mathrm{CTP}\mathcal{W}_m^{\mathbb{C}}(\lambda, \Sigma, L)$. According to Hagedorn et al. [24], the $i$th marginal intensity of $S_k$ follows the $\Gamma(kL, (2\sigma_i^2)^{-1})$ model; the $\mathrm{CTP}\mathcal{W}_m^{\mathbb{C}}$ marginal distribution, say $S_{i,i}$, for the intensity of the $i$th channel has density

$$
f(s_{i,i}) = \sum_{k=1}^{\infty} \Pr(N=k) f_{\Gamma(kL, 2^{-1}\sigma_i^{-2})}(s_{i,i}) = \left[ \frac{e^{-\frac{s_{i,i}}{2\sigma_i^2}}}{s_{i,i}(e^\lambda - 1)} \right] \sum_{k=1}^{\infty} \frac{\left[ \lambda (2^{-1}\sigma_i^{-2}s_{i,i})^L \right]^k}{k! \Gamma(kL)}, \quad (2)
$$

where $\sigma_i^2$ is the $(i,i)$th entry of $\Sigma$.

Now assume that $N \sim \mathrm{Geo}(p)$ and $Z_i \sim \mathcal{W}_m^{\mathbb{C}}(\Sigma, L)$ for $i = 1, \ldots, N$; the coherence matrix per cell following the CGCW model, $S = \sum_{i=1}^{N} Z_i$, has density

$$
f(s) = \sum_{k=1}^{\infty} \Pr(N=k) f_{S_k}(s) = \frac{p e^{-\operatorname{tr}(\Sigma^{-1}s)}}{(1-p)|s|^m} \sum_{k=1}^{\infty} \frac{\left[ (1-p)|\Sigma^{-1}s|^L \right]^k}{\Gamma_m(kL)}.
$$

We denote this situation as $S \sim \mathrm{CG}\mathcal{W}_m^{\mathbb{C}}(p, \Sigma, L)$. The marginal model for intensity of the $i$th channel is:

$$
f(s_{i,i}) = \sum_{k=1}^{\infty} \Pr(N=k) f_{\Gamma(kL, 2^{-1}\sigma_i^{-2})}(s_{i,i}) = \frac{p e^{-\frac{s_{i,i}}{2\sigma_i^2}}}{(1-p)s_{i,i}} \sum_{k=1}^{\infty} \frac{\left[ (1-p)(2^{-1}\sigma_i^{-2}s_{i,i})^L \right]^k}{\Gamma(kL)}. \quad (3)
$$

Figure 2 displays $\mathrm{CTP}\mathcal{W}_m^{\mathbb{C}}$ and $\mathrm{CG}\mathcal{W}_m^{\mathbb{C}}$ marginal densities. It is noticeable that they can be multimodal, an appealing feature of the proposed marginal models for describing high-resolution data, in contrast with other marginals in PolSAR three-parameter laws, e.g., the $\mathcal{G}^0$ and $\mathcal{K}$ laws which are unimodal.

### 2.3. Mathematical Properties

In this section, we derive the cf of $\mathrm{CTP}\mathcal{W}_m^{\mathbb{C}}$ and $\mathrm{CG}\mathcal{W}_m^{\mathbb{C}}$ distributions. To that end, consider the following lemma [31,32].

**Lemma 1.** *Let* $S = \sum\limits_{i=1}^{N} Z_i$ *be such that* $Z_1, \ldots, Z_N$ *is a random sample drawn from* $Z$ *with cf* $\varphi_Z(\cdot)$, *and* $N$ *is a positive integer random variable having cf* $\varphi_N(\cdot)$. *The cf of* $S$, *when* $N$ *and* $Z_i$ *are independent, is given by*

$$
\varphi_S(T) = E(e^{iTS}) = \varphi_N(-i \log \varphi_Z(T)), \text{ where } i = \sqrt{-1}.
$$

The next corollaries hold from Lemma 1.

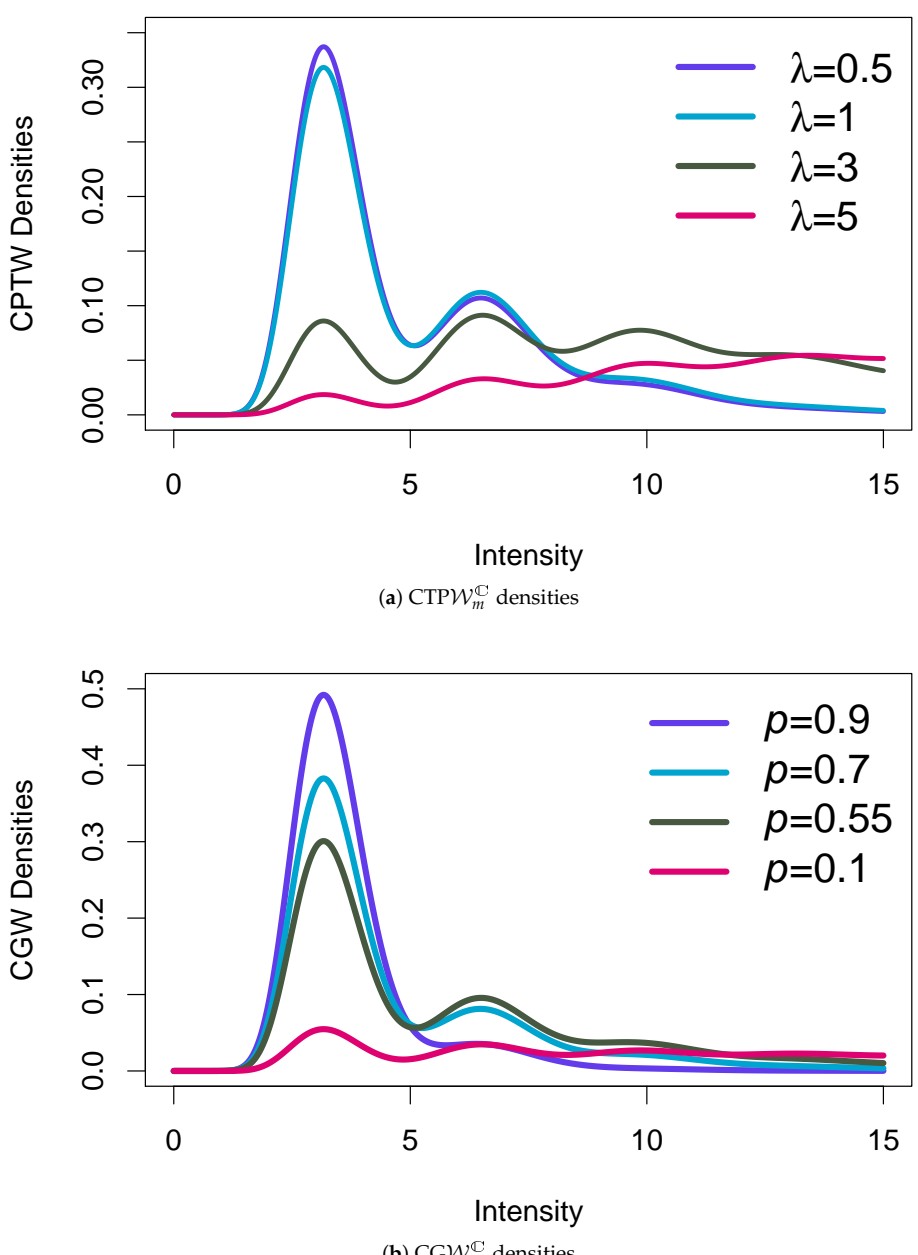

(**a**) CTP$\mathcal{W}_m^{\mathbb{C}}$ densities

(**b**) CG$\mathcal{W}_m^{\mathbb{C}}$ densities

**Figure 2.** Marginal densities CTP$\mathcal{W}_m^{\mathbb{C}}$ and CG$\mathcal{W}_m^{\mathbb{C}}$ distributions.

**Corollary 1.** *Let* $\boldsymbol{Z}_k \sim \mathcal{W}_m^{\mathbb{C}}(L, \boldsymbol{\Sigma})$*, having cf* $\varphi_{\boldsymbol{Z}_k}(\boldsymbol{T}) = E(e^{itr(\boldsymbol{T}\boldsymbol{Z}_k)}) = |\boldsymbol{\Sigma}|^{-L}|\boldsymbol{\Sigma}^{-1} - i\boldsymbol{T}|^{-L}$ *and* $N \sim TPo(\lambda)$ *with* $\varphi_N(t) = \dfrac{e^{\lambda e^{it}} - 1}{e^{\lambda} - 1}$. *Thus, the cf of* $\boldsymbol{S} = \sum_{i=1}^{N} \boldsymbol{Z}_i \sim CTP\mathcal{W}_m^{\mathbb{C}}$ *is*

$$\varphi_{\boldsymbol{S}}(\boldsymbol{T}) = \frac{\exp\left\{ \lambda \exp\left\{ i\left[ -i \log(|\boldsymbol{\Sigma}|^{-L}|\boldsymbol{\Sigma}^{-1} - i\boldsymbol{T}|^{-L}) \right] \right\} \right\} - 1}{e^{\lambda} - 1}$$

$$= \frac{\exp\left\{ \lambda |\boldsymbol{\Sigma}|^{-L}|\boldsymbol{\Sigma}^{-1} - i\boldsymbol{T}|^{-L} \right\} - 1}{e^{\lambda} - 1}.$$

Details about the proof of this corollary (and other theoretical results) are in the Supplementary Materials.

**Corollary 2.** *Let* $Z_i \sim \mathcal{W}_m^{\mathbb{C}}(L, \Sigma)$ *and* $N \sim Geo(p)$ *having cf* $\varphi_N(t) = \dfrac{pe^{it}}{1 - qe^{it}} = \dfrac{p}{e^{-it} - q}$.
*Thus, the cf of* $S = \sum_{i=1}^{N} Z_i \sim CG\mathcal{W}_m^{\mathbb{C}}$ *is expressed by*

$$\varphi_S(T) = \frac{p}{\left(\exp\{-\log(|\Sigma|^{-L}|\Sigma^{-1} - iT|^{-L})\}\right)^{-1} - q} = \frac{p}{\left(|\Sigma|^{L}|\Sigma^{-1} - iT|^{L}\right)^{-1} - q}.$$

With the exception of the complex Wishart law, most PolSAR models have a cf dependent of special functions, e.g., the hypergeometric function, that precludes their manipulation. Corollaries 1 and 2 represent a significant improvement because of their expressiveness and tractability: various moment-kind properties can be derived, e.g., variance and cumulants.

## 3. Maximum Likelihood and Mellin-Based Inference Procedures

This section addresses a two-fold goal: providing an estimation procedure for the proposed models and developing goodness-of-fit (GoF) tools to quantify their adherence to PolSAR data. The first goal is attained with likelihood-based inference, while the second is based on the Mellin transform.

### 3.1. Maximum Likelihood Estimation via EM

In this section, we develop the MLEs for $CTP\mathcal{W}_m^{\mathbb{C}}$ and $CG\mathcal{W}_m^{\mathbb{C}}$ parameters by means of the EM algorithm [33]. Let $S = \sum_{i=1}^{N} Z_i$ be such that $N \sim \{TP(\lambda) \text{ or } Geo(p)\}$ and $Z_i \sim \mathcal{W}_m^{\mathbb{C}}(\Sigma, L)$ for $i = 1, \ldots, N$ and both pairs $(Z_i, Z_j)$ and $(N, Z_i)$ are independent $\forall i \neq j$. Consider that $S_1, \ldots, S_T$ is a $T$-points random sample drawn from $S \sim \{CTP\mathcal{W}_m^{\mathbb{C}}, CG\mathcal{W}_m^{\mathbb{C}}\}$. Note that $S = (S_1, \ldots, S_T)$ is observable, but $N = (N_1, \ldots, N_T)$ is not. Let $(S, N)$ be the complete set of observations having joint density (for $\theta = (\lambda, \text{vec}(\Sigma)^{\top})^{\top}$ or $(p, \text{vec}(\Sigma)^{\top})^{\top}$), then

$$L^c(\theta \mid n, s) = f_{S,N}(s, n \mid \theta) = \prod_{i=1}^{T} f_{S_i, N_i}(s_i, n_i \mid \theta) = \prod_{i=1}^{T} f_{S_i \mid N_i = n_i}(s_i \mid n_i, \theta) \Pr(N_i = n_i),$$

where $s = (s_1, \ldots, s_T)$ and $n = (n_1, \ldots, n_T)$ are outcomes of $S$ and $N$. Thus, the MLEs via EM are determined by the next steps:

- Step E: Derive

$$Q(\theta \mid \theta_0, s) := E_{\theta_0}[\log L^c(\theta \mid N, s)],$$

where $E_{\theta_0}$ is the expected value with respect to $[N \mid \theta_0, s]$ wih pmf $f(N \mid \theta, s)$.
- Step M: In the $(t+1)$th iteration, find $\hat{\theta}^{(t+1)}$ that maximizes $Q(\theta \mid \theta^{(t)}, s)$,

$$\hat{\theta}^{(t+1)} = \arg\max_{\theta \in \Theta} Q(\theta \mid \theta_0^{(t)}, s),$$

where $\Theta$ is the parametric space.

These steps should be repeated until convergence is achieved. To this aim, we adopt as a stopping criterion $\|\theta^{(t+1)} - \theta^{(t)}\| < \epsilon$, where $\|\cdot\|$ is the Euclidean norm function and $\epsilon$ is a specified precision level, defined as $\epsilon = 1 \times 10^{-4}$ in this study. The next corollary determines MLE expressions.

**Corollary 3.** *For the* $CTP\mathcal{W}_m^{\mathbb{C}}$ *distribution, the MLE for* $\theta = (\lambda, vec(\Sigma)^{\top})^{\top}$ *is* $\hat{\theta} = (\hat{\lambda}, vec(\hat{\Sigma})^{\top})^{\top}$
*where* $\hat{\lambda}$ *is defined as a root of the nonlinear equation,*

$$\frac{\hat{\lambda}^{(t+1)}}{(1 - e^{-\hat{\lambda}^{(t+1)}})} = \frac{1}{T} \sum_{i=1}^{T} E(N \mid \theta^{(t)}, S = s_i) \tag{4}$$

*and*

$$\hat{\boldsymbol{\Sigma}}^{(t+1)} = \frac{T\overline{\boldsymbol{S}}}{L \sum_{i=1}^{T} E(N \mid \boldsymbol{\theta}^{(t)}, \boldsymbol{S} = \boldsymbol{s}_i)}, \tag{5}$$

*where* $\overline{\boldsymbol{S}} = T^{-1} \sum_{i=1}^{T} \boldsymbol{s}_i$, *and*

$$
\begin{aligned}
E(N \mid \boldsymbol{\theta}^{(t)}, \boldsymbol{S} = \boldsymbol{s}_i) &= \sum_{i=1}^{\infty} k_i P(N = k_i \mid \boldsymbol{\theta}^{(t)}, \boldsymbol{S} = \boldsymbol{s}_i) \\
&= \sum_{i=1}^{\infty} k_i \frac{f_{S_{k_i}}(\boldsymbol{s}_i \mid \boldsymbol{\theta}^{(t)}) \cdot P(N = k_i \mid \boldsymbol{\theta}^{(t)})}{f_S(\boldsymbol{s}_i \mid \boldsymbol{\theta}^{(t)})} \\
&= \frac{\sum_{i=1}^{\infty} k_i \left[ \left( \frac{\lambda^{(t)\frac{1}{L}} \mid \boldsymbol{s}_i \mid}{|\boldsymbol{\Sigma}^{(t)}|} \right)^L \right]^{k_i} \frac{1}{\Gamma_m(k_i L)}}{\sum_{j=1}^{\infty} \left[ \left( \frac{\lambda^{(t)\frac{1}{L}} \mid \boldsymbol{s}_i \mid}{|\boldsymbol{\Sigma}^{(t)}|} \right)^L \right]^{j} \frac{1}{\Gamma_m(jL)}}.
\end{aligned}
$$

*For the* $CG\mathcal{W}_m^{\mathbb{C}}$ *distribution, the MLE for* $\boldsymbol{\theta} = (p, vec(\boldsymbol{\Sigma})^{\top})^{\top}$ *is* $\hat{\boldsymbol{\theta}} = (\hat{p}, vec(\hat{\boldsymbol{\Sigma}})^{\top})^{\top}$, *where*

$$\hat{p}^{(t+1)} = \frac{T}{\sum_{i=1}^{T} E(N \mid \boldsymbol{\theta}^{(t)}, \boldsymbol{S} = \boldsymbol{s}_i)}, \tag{6}$$

*and*

$$\hat{\boldsymbol{\Sigma}}^{(t+1)} = \frac{T\overline{\boldsymbol{S}}}{L \sum_{i=1}^{T} E(N \mid \boldsymbol{\theta}^{(t)}, \boldsymbol{S} = \boldsymbol{s}_i)}, \tag{7}$$

*where*

$$
\begin{aligned}
E(N \mid \boldsymbol{\theta}^{(t)}, \boldsymbol{S} = \boldsymbol{s}_i) &= \sum_{i=1}^{\infty} k_i P(N = k_i \mid \boldsymbol{\theta}^{(t)}, \boldsymbol{S} = \boldsymbol{s}) \\
&= \sum_{i=1}^{\infty} k_i \frac{f_{S_{k_i}}(\boldsymbol{s}_i \mid \boldsymbol{\theta}^{(t)}) \cdot P(N = k_i \mid \boldsymbol{\theta}^{(t)})}{f_S(\boldsymbol{s}_i \mid \boldsymbol{\theta}^{(t)})} \\
&= \frac{\sum_{i=1}^{\infty} k_i \left[ \left( \frac{q^{\frac{1}{L}} |\boldsymbol{s}_i|}{|\boldsymbol{\Sigma}^{(t)}|} \right)^L \right]^{k_i} \frac{1}{\Gamma_m(k_i L)}}{\sum_{j=1}^{\infty} \left[ \left( \frac{q^{\frac{1}{L}} \mid \boldsymbol{s}_i \mid}{|\boldsymbol{\Sigma}^{(t)}|} \right)^L \right]^{j} \frac{1}{\Gamma_m(jL)}},
\end{aligned}
$$

*and* $q = 1 - p_0$.

### 3.2. Mellin Diagram

The Mellin-kind transform applied to hermitian random matrices is defined on $\boldsymbol{\Omega}_+$ [19] and, therefore, can be applied to the $CTP\mathcal{W}_m^{\mathbb{C}}$ and $CG\mathcal{W}_m^{\mathbb{C}}$ models. Anfinsen and Eltoft [10] have shown that this transform may be used as a new statistical inference mechanism by means of Mellin-kind statistic (MKS). MKSs have been derived for the $s\mathcal{W}_m^{\mathbb{C}}$ [23], $\mathcal{K}_m$ [18], $\mathcal{G}_m^0$ [17], and $\mathcal{U}_m$ [16] distributions, along with GoF tools having graphical appeal. In what follows, this paper tackles the derivation of the $CTP\mathcal{W}_m^{\mathbb{C}}$ and $CG\mathcal{W}_m^{\mathbb{C}}$ MKSs in order to evaluate the fitting quality of proposed models.

Let $C \in \Omega_+$, then the complex matrix-variate Mellin-kind transform of a real-valued function, say $g(C) \colon \Omega_+ \to \mathbb{R}$, is

$$\phi_C(s) = \mathcal{M}\{g(C)\}(s) = \int_{\Omega_+} |C|^{s-m} g(C) dC,$$

with $s \in \mathbb{C}$, whenever the integral exists. Let $g(C) = \sum_{k=1}^{\infty} P(N=k) f_{S_k}(C)$ such that

$$f_{S_k}(C) = \frac{|C|^{(kL)-m}}{|\Sigma|^{(kL)} \Gamma_m(kL)} \exp\left\{-\text{tr}(\Sigma^{-1} C)\right\}.$$

Thus,

$$\phi_C(s) = \mathcal{M}\{g(C)\}(s) = \int_{\Omega_+} \sum_{k=1}^{\infty} \text{Pr}(N=k) \frac{|C|^{(kL)+s-2m}}{|\Sigma|^{(kL)} \Gamma_m(kL)} e^{-\text{tr}(\Sigma^{-1} C)} dC.$$

After some algebraic manipulations,

$$\begin{aligned}
\phi_C(s) &= \sum_{k=1}^{\infty} \text{Pr}(N=k) \frac{|\Sigma|^{(kL+s-m)} \Gamma_m(kL+s-m)}{|\Sigma|^{(kL)} \Gamma_m(kL)} \\
&\quad \int_{\Omega_+} \frac{|C|^{(kL)+s-2m}}{|\Sigma|^{(kL+s-m)} \Gamma_m(kL+s-m)} e^{-\text{tr}(\Sigma^{-1} C)} dC \\
&= \sum_{k=1}^{\infty} \text{Pr}(N=k) |\Sigma|^{(s-m)} \frac{\Gamma_m(kL+s-m)}{\Gamma_m(kL)}.
\end{aligned} \tag{8}$$

The Mellin-kind cumulant-generating (MCGF) function is defined as:

$$\varphi_C(s) = \log \phi_C(s),$$

and the $\nu$th-order MLC is defined as

$$\kappa_\nu C = \frac{d^\nu}{ds^\nu} \varphi_C(s) \bigg|_{s=m}.$$

1.  Thus, for $N \sim \text{TPo}(\lambda)$ in (8), MCGF is given by

$$\varphi_C(s) = -\log(e^\lambda - 1) + (s-m)\log|\Sigma| + \log\left[\sum_{k=1}^{\infty} \frac{\lambda^k}{k!} \frac{\Gamma_m(kL+s-m)}{\Gamma_m(kL)}\right],$$

and
2.  for $N \sim \text{Geo}(p)$ in (8), MCGF has the form

$$\varphi_C(s) = -\log p + (s-m)\log|\Sigma| + \log\left[\sum_{k=1}^{\infty} q^{(k-1)} \frac{\Gamma_m(kL+s-m)}{\Gamma_m(kL)}\right].$$

Three $\nu$th-order MLCs for $\text{CTP}\mathcal{W}_m^{\mathbb{C}}$ and $\text{CG}\mathcal{W}_m^{\mathbb{C}}$ laws are given in the next proposition.

**Corollary 4.** *The first, second, and third MLCs for the CTP$\mathcal{W}_m^{\mathbb{C}}$ and CG$\mathcal{W}_m^{\mathbb{C}}$ models are expressed by, respectively,*

$$\kappa_1 = \log|\boldsymbol{\Sigma}| + \left( \sum_{k=1}^{\infty} \Pr(N = k)\psi_m^{(0)}(kL) \right),$$

$$\kappa_2 = -\left[ \sum_{k=1}^{\infty} \Pr(N = k)\psi_m^{(0)}(kL) \right]^2 + \sum_{k=1}^{\infty} \Pr(N = k)\left[ \left(\psi_m^{(0)}(kL)\right)^2 + \psi_m^{(1)}(kL) \right]$$

*and*

$$\kappa_3 = 2\left( \sum_{k=1}^{\infty} \Pr(N = k)\psi_m^{(0)}(kL) \right)^3 - 3\left( \sum_{k=1}^{\infty} \Pr(N = k)\psi_m^{(0)}(kL) \right)$$

$$\times \left\{ \sum_{k=1}^{\infty} \Pr(N = k)\left[ \left(\psi_m^{(0)}(kL)\right)^2 + \psi_m^{(1)}(kL) \right] \right\}$$

$$+ \sum_{k=1}^{\infty} \Pr(N = k)\left[ \left(\psi_m^{(0)}(kL)\right)^3 + 3\psi_m^{(0)}(kL)\psi_m^{(1)}(kL) + \psi_m^{(2)}(kL) \right].$$

Based on these results, we can construct GoF tools for assessing our proposals in practice. To develop the Mellin diagram, one should first estimate the parameters and determine a quota for summations of proposed models. Next, the parametric cumulants $\kappa_2$ and $\kappa_3$ can be obtained and a curve is obtained for each distribution. Thus, from sample cumulants based on PolSAR data, it is possible to check the adherence of the proposed model to the data.

## 4. Results and Discussion

### 4.1. Analysis of Simulated Data

We devised a Monte Carlo experiment to quantify the asymptotic behavior of MLEs for parameters $[\lambda, \boldsymbol{\Sigma}]$ and $[p, \boldsymbol{\Sigma}]$. To this end, CTP$\mathcal{W}_m^{\mathbb{C}}$ and CG$\mathcal{W}_m^{\mathbb{C}}$ samples were generated having sizes $T = 10, 30, 100, 1000$. We produced one thousand replicas for each. We assumed the number of looks $L = 4$ and, as common matrix,

$$\boldsymbol{\Sigma} = \begin{pmatrix} 0.07582 & 0.00364 + 0.00388i & 0.01604 + 0.01125i \\ 0.00364 - 0.00388i & 0.03737 & 0.00151 + 0.00202i \\ 0.01604 - 0.01125i & 0.00151 - 0.00202i & 0.06308 \end{pmatrix},$$

with ($|\boldsymbol{\Sigma}| = 0.00016$ and $\text{tr}(\boldsymbol{\Sigma}) = 0.17626$). This matrix was determined by the average value of returns at the area AS in Figure 3 (the Foulum image discussed in detail in the Section 4.2). Notice that this is a heterogeneous area with observations coming from more than a single distribution.

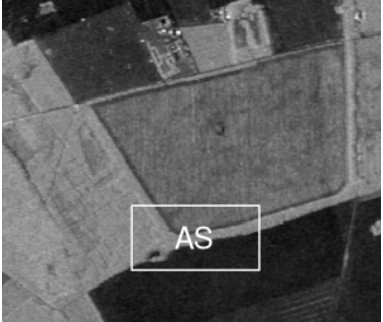

**Figure 3.** Selected area of EMISAR image Foullum (channel HH) to determine the average matrix as a real values form in the data simulation.

For CTP$\mathcal{W}_m^{\mathbb{C}}$ and CG$\mathcal{W}_m^{\mathbb{C}}$ models, values of $\lambda \in \{0.1, 0.5, 1.0\}$ and $p \in \{0.99, 0.7, 0.4\}$ were chosen to represent returned matrices from homogeneous scenes to heterogeneous ones. We used the mean square error and bias as figures of merit to assess the performance of the estimates.

Tables 2 and 3 display estimated parameters under the CTP$\mathcal{W}_m^{\mathbb{C}}$ and CG$\mathcal{W}_m^{\mathbb{C}}$ models, respectively. In general, we observe the smallest bias and MSE with the largest sample sizes, as expected. More pronounced texture targets (as for $\lambda = 1$ and $p = 0.4$) required larger sample sizes for achieving better quality measures than in homogeneous situations.

**Table 2.** Performance of MLEs for CTP$\mathcal{W}_m^{\mathbb{C}}$ data.

| $n$ | $\hat{\lambda}$ $\left(MSE_{\hat{\lambda}}\right)$ | $\mathrm{tr}(\hat{\Sigma})$ $\left(MSE_{\mathrm{tr}(\hat{\Sigma})}\right)$ | $\|\hat{\Sigma}\|$ $\left(MSE_{\|\hat{\Sigma}\|}\right)$ |
|---|---|---|---|
| | | $\lambda = 0.10$ | |
| 10 | 0.1288 | 0.17321 | 0.00014 |
| | (0.04166) | $(4.00 \times 10^{-4})$ | $(2.49 \times 10^{-9})$ |
| 30 | 0.1087 | 0.17544 | 0.00015 |
| | (0.01073) | $(1.20 \times 10^{-4})$ | $(8.25 \times 10^{-10})$ |
| 100 | 0.1011 | 0.17608 | 0.00016 |
| | (0.00298) | $(3.78 \times 10^{-5})$ | $(2.54 \times 10^{-10})$ |
| 1000 | 0.0999 | 0.17627 | 0.00016 |
| | (0.00029) | $(3.54 \times 10^{-6})$ | $(2.41 \times 10^{-11})$ |
| | | $\lambda = 0.50$ | |
| 10 | 0.5644 | 0.17140 | 0.00014 |
| | (0.14842) | $(5.30 \times 10^{-4})$ | $(3.52 \times 10^{-9})$ |
| 30 | 0.5197 | 0.17553 | 0.00015 |
| | (0.03998) | $(1.40 \times 10^{-4})$ | $(1.01 \times 10^{-9})$ |
| 100 | 0.5017 | 0.17589 | 0.00016 |
| | (0.01160) | $(4.06 \times 10^{-5})$ | $(2.89 \times 10^{-10})$ |
| 1000 | 0.49931 | 0.1763 | 0.00016 |
| | (0.00112) | $(3.95 \times 10^{-6})$ | $(2.87 \times 10^{-11})$ |
| | | $\lambda = 1.00$ | |
| 10 | 1.0745 | 0.17088 | 0.00015 |
| | (0.26535) | $(6.70 \times 10^{-4})$ | $(4.95 \times 10^{-9})$ |
| 30 | 1.0115 | 0.17549 | 0.00016 |
| | (0.06560) | $(1.80 \times 10^{-4})$ | $(1.36 \times 10^{-9})$ |
| 100 | 0.9923 | 0.17710 | 0.00016 |
| | (0.02004) | $(5.19 \times 10^{-5})$ | $(3.94 \times 10^{-10})$ |
| 1000 | 0.9834 | 0.17751 | 0.00016 |
| | (0.00228) | $(7.09 \times 10^{-6})$ | $(5.42 \times 10^{-11})$ |

**Table 3.** Performance of MLEs for CG$\mathcal{W}_m^{\mathbb{C}}$ data.

| $n$ | $\hat{p}$ $\left(MSE_{\hat{p}}\right)$ | $\mathrm{tr}(\hat{\Sigma})$ $\left(MSE_{\mathrm{tr}(\hat{\Sigma})}\right)$ | $\|\hat{\Sigma}\|$ $\left(MSE_{\|\hat{\Sigma}\|}\right)$ |
|---|---|---|---|
| | | $p = 0.40$ | |
| 10 | 0.4903 | 0.21160 | 0.00035 |
| | (0.02018) | (0.00506) | $(2.47 \times 10^{-7})$ |
| 30 | 0.5215 | 0.22931 | 0.00040 |
| | (0.02096) | (0.00483) | $(1.16 \times 10^{-7})$ |
| 100 | 0.5256 | 0.23184 | 0.00038 |
| | (0.01760) | (0.00366) | $(6.19 \times 10^{-8})$ |
| 1000 | 0.5310 | 0.23397 | 0.00038 |
| | (0.01733) | (0.00338) | $(4.89 \times 10^{-8})$ |
| | | $p = 0.70$ | |
| 10 | 0.7183 | 0.17387 | 0.00015 |
| | (0.02056) | (0.00054) | $(4.23 \times 10^{-9})$ |
| 30 | 0.7056 | 0.17607 | 0.00016 |
| | (0.00616) | (0.00018) | $(1.41 \times 10^{-9})$ |
| 100 | 0.7095 | 0.17756 | 0.00017 |
| | (0.00176) | $(7.00 \times 10^{-5})$ | $(5.53 \times 10^{-10})$ |
| 1000 | 0.7070 | 0.17800 | 0.00017 |
| | (0.00022) | $(8.75 \times 10^{-6})$ | $(6.62 \times 10^{-11})$ |
| | | $p = 0.99$ | |
| 10 | 0.9828 | 0.17450 | 0.00015 |
| | (0.00235) | (0.00033) | $(2.17 \times 10^{-9})$ |
| 30 | 0.9878 | 0.17646 | 0.00016 |
| | (0.00057) | (0.00011) | $(7.55 \times 10^{-10})$ |
| 100 | 0.9900 | 0.17570 | 0.00016 |
| | (0.00017) | $(3.32 \times 10^{-5})$ | $(2.09 \times 10^{-10})$ |
| 1000 | 0.9909 | 0.17640 | 0.00016 |
| | $(2.22 \times 10^{-5})$ | $(3.29 \times 10^{-6})$ | $(2.27 \times 10^{-11})$ |

### 4.2. Analysis of Data from Actual Sensors

Firstly, we studied an AIRSAR (Airborne Synthetic Aperture Radar) image of a region of the San Francisco bay (USA). This image was captured with four nominal looks. Secondly, we applied our proposals to an EMISAR (SAR image system of the Electromagnetics Institute) image of the Foullum (DK) region that has eight nominal looks. Thirdly, we analysed an E-SAR image of the Neubrandenburg (Northeastern Germany) agricultural areas, obtained at during the AGRISAR flight campaign and with ten nominal looks.

In order to illustrate the effect of the texture over the values of MLEs, Figure 4 displays the San Francisco image and three of its highlighted areas in the HH channel. Areas A1, A2, and A3 represent ocean (least textured case), forest (intermediate texture), and urban (strongly textured) scenarios, respectively. Table 4 presents MLEs for $s\mathcal{W}_m^{\mathbb{C}}$, $\text{CTP}\mathcal{W}_m^{\mathbb{C}}$ and $\text{CG}\mathcal{W}_m^{\mathbb{C}}$ parameters in regions A1, A2, and A3, and in the full image. For MLEs of $\boldsymbol{\Sigma}$, the $\text{CTP}\mathcal{W}_m^{\mathbb{C}}$ model presented a performance closer to $s\mathcal{W}_m^{\mathbb{C}}$ than $\text{CG}\mathcal{W}_m^{\mathbb{C}}$, mainly in ocean scenarios (for which the literature [17,34] suggests using $s\mathcal{W}_m^{\mathbb{C}}$). With respect to additional parameters $\lambda$ and $p$, large values of $\hat{\lambda}$ or small values of $\hat{p}$ indicate regions with more pronounced textures; while $(\lambda, p) \longrightarrow (0, 1)$ indicates that the new models collapse in $s\mathcal{W}_m^{\mathbb{C}}$.

Now we estimate the parameters of both distributions for the whole image over non-overlapping windows of size $7 \times 7 = 49$ pixels. Figures 5–7 show maps of these estimates for San Francisco, Foulum and Neubrandenburg images, respectively.

**Table 4.** MLE for the $s\mathcal{W}_m^{\mathbb{C}}(L, \boldsymbol{\Sigma})$, $\text{CTP}\mathcal{W}_m^{\mathbb{C}}(\lambda, L, \boldsymbol{\Sigma})$, and $\text{CG}\mathcal{W}_m^{\mathbb{C}}(p, L, \boldsymbol{\Sigma})$ distributions with $L = 4$.

|  | Model | $|\hat{\boldsymbol{\Sigma}}|$ | $\hat{\lambda}$ | $\hat{p}$ | $k$ |
|---|---|---|---|---|---|
| A1 | $s\mathcal{W}_m^{\mathbb{C}}$ | $2.39 \times 10^{-9}$ | • | • | • |
| | $\text{CTP}\mathcal{W}_m^{\mathbb{C}}$ | $2.39 \times 10^{-9}$ | $1.00 \times 10^{-5}$ | • | 4 |
| | $\text{CG}\mathcal{W}_m^{\mathbb{C}}$ | $1.94 \times 10^{-9}$ | • | 0.93 | 4 |
| A2 | $s\mathcal{W}_m^{\mathbb{C}}$ | $5.38 \times 10^{-7}$ | • | • | • |
| | $\text{CTP}\mathcal{W}_m^{\mathbb{C}}$ | $5.38 \times 10^{-7}$ | $9.73 \times 10^{-5}$ | • | 6 |
| | $\text{CG}\mathcal{W}_m^{\mathbb{C}}$ | $3.81 \times 10^{-7}$ | • | 0.89 | 9 |
| A3 | $s\mathcal{W}_m^{\mathbb{C}}$ | $1.90 \times 10^{-5}$ | • | • | • |
| | $\text{CTP}\mathcal{W}_m^{\mathbb{C}}$ | $1.37 \times 10^{-5}$ | 0.22 | • | 5 |
| | $\text{CG}\mathcal{W}_m^{\mathbb{C}}$ | $1.19 \times 10^{-5}$ | • | 0.86 | 5 |
| full image | $s\mathcal{W}_m^{\mathbb{C}}$ | $1.18 \times 10^{-5}$ | • | • | • |
| | $\text{CTP}\mathcal{W}_m^{\mathbb{C}}$ | $8.00 \times 10^{-6}$ | 0.26 | • | 6 |
| | $\text{CG}\mathcal{W}_m^{\mathbb{C}}$ | $7.29 \times 10^{-6}$ | • | 0.85 | 6 |

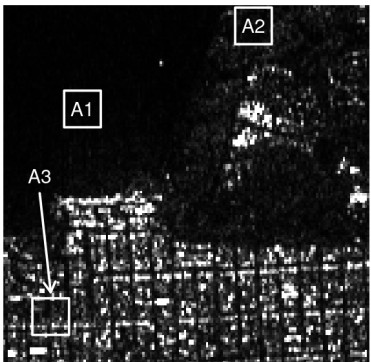

**Figure 4.** AIRSAR image of San Francisco (channel HH) with selected regions.

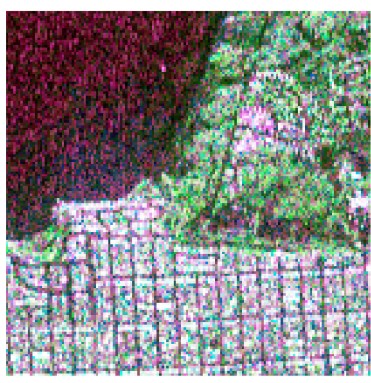

(**a**) Pauli representation for San Francisco Image.

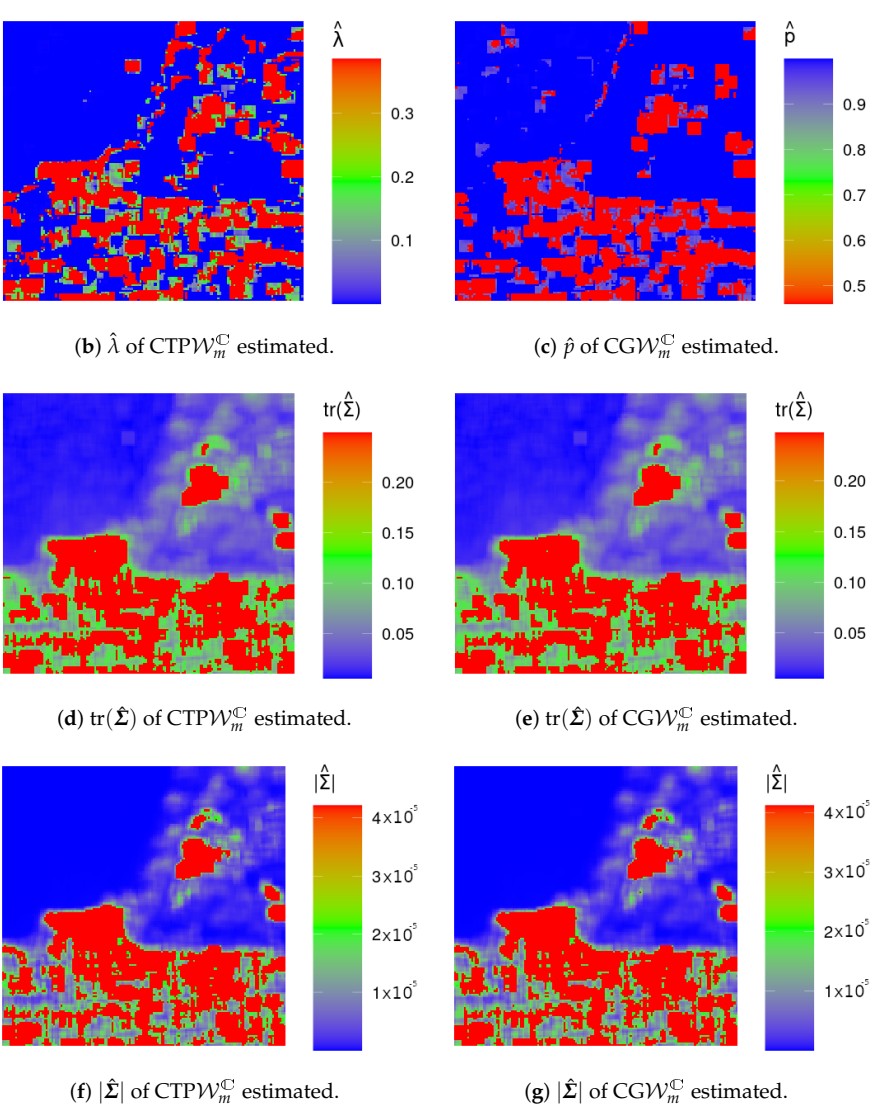

**Figure 5.** Maps of estimated parameters of San Francisco for CTP$\mathcal{W}_m^{\mathbb{C}}$ (**left**) and CG$\mathcal{W}_m^{\mathbb{C}}$ (**right**) distributions, respectively.

MLE estimates for $\lambda$, tr($\Sigma$), and $|\Sigma|$ under CTP$\mathcal{W}_m^{\mathbb{C}}$ are exhibited in Figure 5b,d,f, respectively. Figure 5c,e,g display the maps of MLEs of $p$, tr($\Sigma$), and $|\Sigma|$, respectively, under the CG$\mathcal{W}_m^{\mathbb{C}}$ model.

The pairs of Figure 5d,e and Figure 5f,g show very similar results. The trace and the determinant are able to identify three regions: urban areas appear in green, forests in orange, and ocean in red. In Figure 5b,c, the MLEs for $\lambda$ and $p$ assumed values in $(0,1)$ and $(0.5,1)$, respectively. It is noticeable that the subsets $[p < 0.6]$ and $[\lambda > 0.3]$ indicate urban scenarios, under the hypothesis that AIRSAR return follows $\mathrm{CPT}\mathcal{W}_m^{\mathbb{C}}$ and $\mathrm{CG}\mathcal{W}_m^{\mathbb{C}}$ models.

Figure 6 shows values of MLEs for the Foulum image. Maps for $\hat{\lambda}$ and $\hat{p}$ are in Figure 6b,c. Edges and background areas of the image are highlighted. Figure 6d,e presents the estimates of $\mathrm{tr}(\hat{\boldsymbol{\Sigma}})$ and $|\hat{\boldsymbol{\Sigma}}|$. The largest estimates addressed areas of Conifer, Wheat and Rapeseed, while the smallest values were associated with other areas.

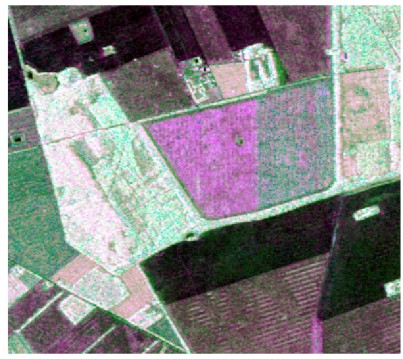

(**a**) Pauli representation for Foulum Image.

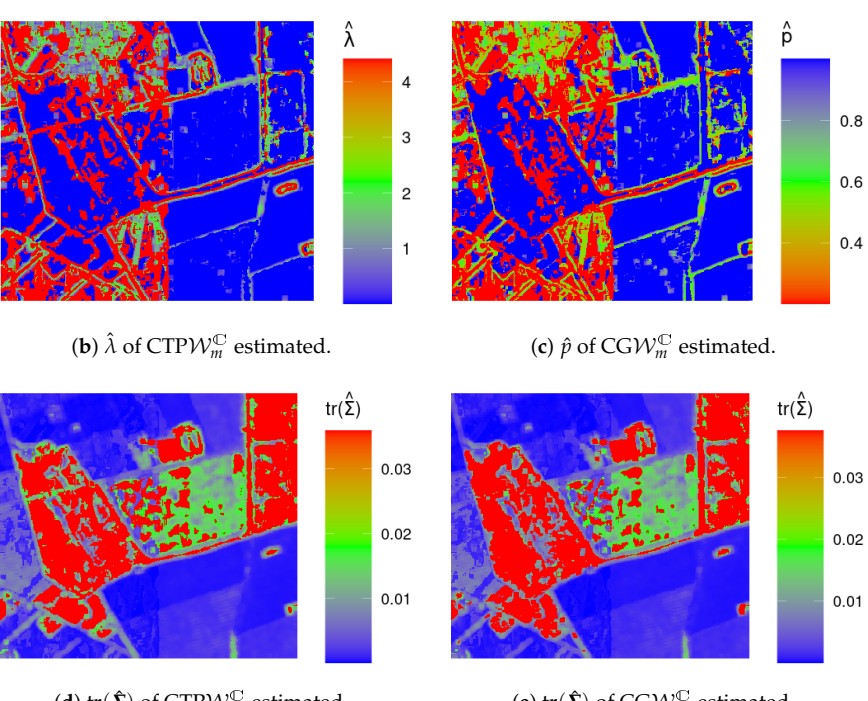

(**b**) $\hat{\lambda}$ of $\mathrm{CTP}\mathcal{W}_m^{\mathbb{C}}$ estimated.

(**c**) $\hat{p}$ of $\mathrm{CG}\mathcal{W}_m^{\mathbb{C}}$ estimated.

(**d**) $\mathrm{tr}(\hat{\boldsymbol{\Sigma}})$ of $\mathrm{CTP}\mathcal{W}_m^{\mathbb{C}}$ estimated.

(**e**) $\mathrm{tr}(\hat{\boldsymbol{\Sigma}})$ of $\mathrm{CG}\mathcal{W}_m^{\mathbb{C}}$ estimated.

**Figure 6.** *Cont.*

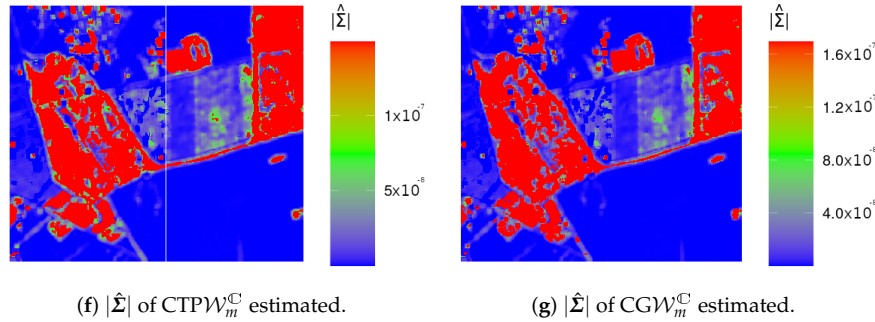

(**f**) $|\hat{\boldsymbol{\Sigma}}|$ of CTP$\mathcal{W}_m^{\mathbb{C}}$ estimated.

(**g**) $|\hat{\boldsymbol{\Sigma}}|$ of CG$\mathcal{W}_m^{\mathbb{C}}$ estimated.

**Figure 6.** Maps of estimated parameters of Foulum for CTP$\mathcal{W}_m^{\mathbb{C}}$ (**left**) and CG$\mathcal{W}_m^{\mathbb{C}}$ (**right**) distributions, respectively.

Maps of MLEs for the Neubrandenburg image are exhibited in Figure 7. From both Figure 7b,c and the AgriSAR 2006 report (https://earth.esa.int/eogateway/campaigns/agrisar-2006, accessed on 5 September 2022), large values of $\hat{\lambda}$ identify high levels of roughness, and built areas from crop ones [35]. In Figure 7b–g, it is possible to identify five kinds of crops: winter rape (yellow), winter wheat (red), maize (green), winter barley (light red) and sugar beet (light green).

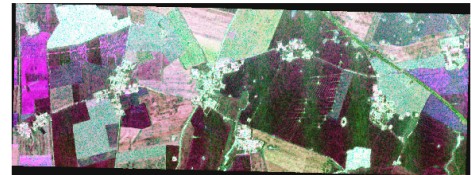

(**a**) Pauli representation for Demmin - Görmin Image.

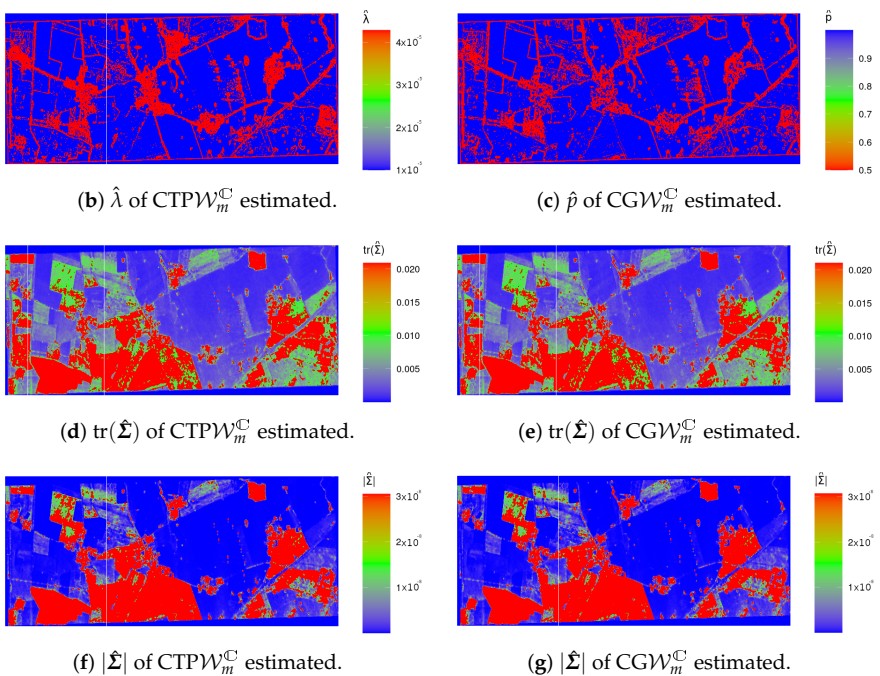

(**b**) $\hat{\lambda}$ of CTP$\mathcal{W}_m^{\mathbb{C}}$ estimated.

(**c**) $\hat{p}$ of CG$\mathcal{W}_m^{\mathbb{C}}$ estimated.

(**d**) tr$(\hat{\boldsymbol{\Sigma}})$ of CTP$\mathcal{W}_m^{\mathbb{C}}$ estimated.

(**e**) tr$(\hat{\boldsymbol{\Sigma}})$ of CG$\mathcal{W}_m^{\mathbb{C}}$ estimated.

(**f**) $|\hat{\boldsymbol{\Sigma}}|$ of CTP$\mathcal{W}_m^{\mathbb{C}}$ estimated.

(**g**) $|\hat{\boldsymbol{\Sigma}}|$ of CG$\mathcal{W}_m^{\mathbb{C}}$ estimated.

**Figure 7.** Maps of estimated parameters of CTP$\mathcal{W}_m^{\mathbb{C}}$ (**left**) and CG$\mathcal{W}_m^{\mathbb{C}}$ (**right**) distributions, respectively.

Now, we are in position to describe actual PolSAR scenarios by means of CTPC$\mathcal{W}_m^{\mathbb{C}}$ and CGC$\mathcal{W}_m^{\mathbb{C}}$ distributions, comparatively to three well-defined models: s$\mathcal{W}_m^{\mathbb{C}}$, $\mathcal{K}_m$, and $\mathcal{G}_m^0$. Three scenes of San Francisco and six of Foulum to represent one area type are displayed

in Figures 8a and 9a, respectively. In this case, a windows of $16 \times 16 = 256$ pixels was considered to represent each region of the image. In these windows, 100 random samples with replacements were made with a length of 128 pixels and the average was extracted [10]. Figures 8c and 9b exhibit the MLC diagram having: (i) the projection curves due to five considered matrix models and (ii) pairs of sample MLCs ($\kappa_2$, $\kappa_3$) of highlighted regions in Figures 8a and 9a. The sample MLCs of each image sample have been plotted over the population MLC manifolds of the s$\mathcal{W}_m^{\mathbb{C}}$, $\mathcal{K}_m$, $\mathcal{G}_m^0$, CTP$\mathcal{W}_m^{\mathbb{C}}$ and CG$\mathcal{W}_m^{\mathbb{C}}$ distributions.

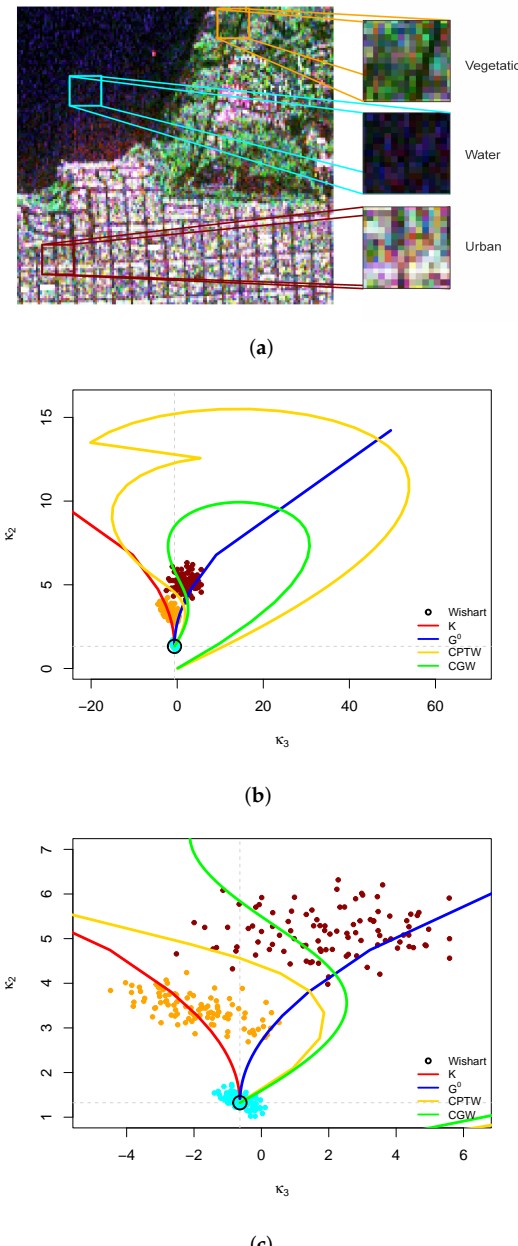

**Figure 8.** MLC diagram with sample MLCs computed from the samples of San Francisco. (**a**) Subset areas of San Francisco. (**b**) MLC diagram with sample MLCs computed from the samples in (**a**). (**c**) Zoom on the data.

In Figure 8a, we have extracted ocean (cyan square), vegetation (gold square), and urban (dark red square) samples. According to the MLC maps, the ocean sample had the best fit at the s$\mathcal{W}_m^{\mathbb{C}}$ distribution, the vegetation one overlapped the $\mathcal{K}_m$ curve and the urban sample assumed the best fit on three curves: $\mathcal{G}_m^0$, CTP$\mathcal{W}_m^{\mathbb{C}}$ (with larger number of points)

and CGW$_m^{\mathbb{C}}$. It is worth highlighting that there is a continuity break in the CTPW$_m^{\mathbb{C}}$ $(\kappa_2, \kappa_3)$ curve for this case.

In Figure 9a, we have extracted the background (green square; representing unknown areas), Rapeseed (pink square), Wheat (cyan square), Oat (orange square), Rye (gray square), and Conifer (teal square) samples. The Conifer, Rye, and Wheat samples have overlapped on the sW$_m^{\mathbb{C}}$ curve. The Rapeseed sample fitted on the $\mathcal{G}_m^0$ distribution, the background sample had the points between the $\mathcal{G}_m^0$ and CGW$_m^{\mathbb{C}}$ laws, and the Oat sample had the best fit on the CTPW$_m^{\mathbb{C}}$ curve.

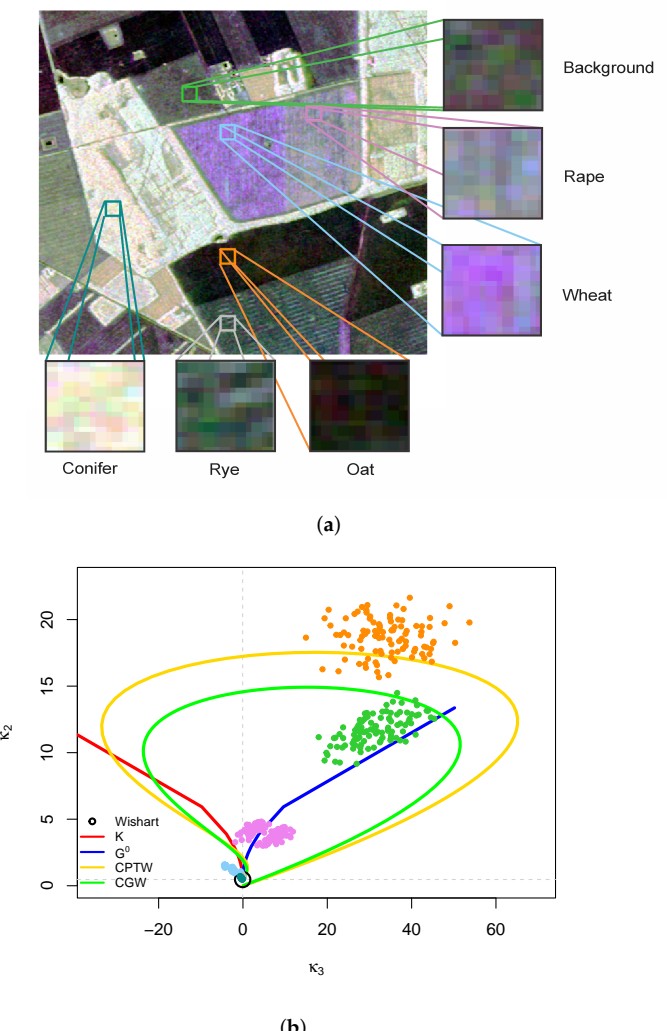

(**a**)

(**b**)

**Figure 9.** MLC diagram with sample MLCs computed from the samples of Foulum. (**a**) Subset areas of Foulum [36]. (**b**) MLC diagram with sample MLCs computed from the samples in (**a**).

Figure 10c highlights five crop scenes: winter barley (purple), sugar beet (gray), winter wheat (dark green), winter rape (brown) and maize (light blue). The sW$_m^{\mathbb{C}}$ distribution presented the best fit for three last scenes. The two proposed models furnished the best description for Winter Barley.

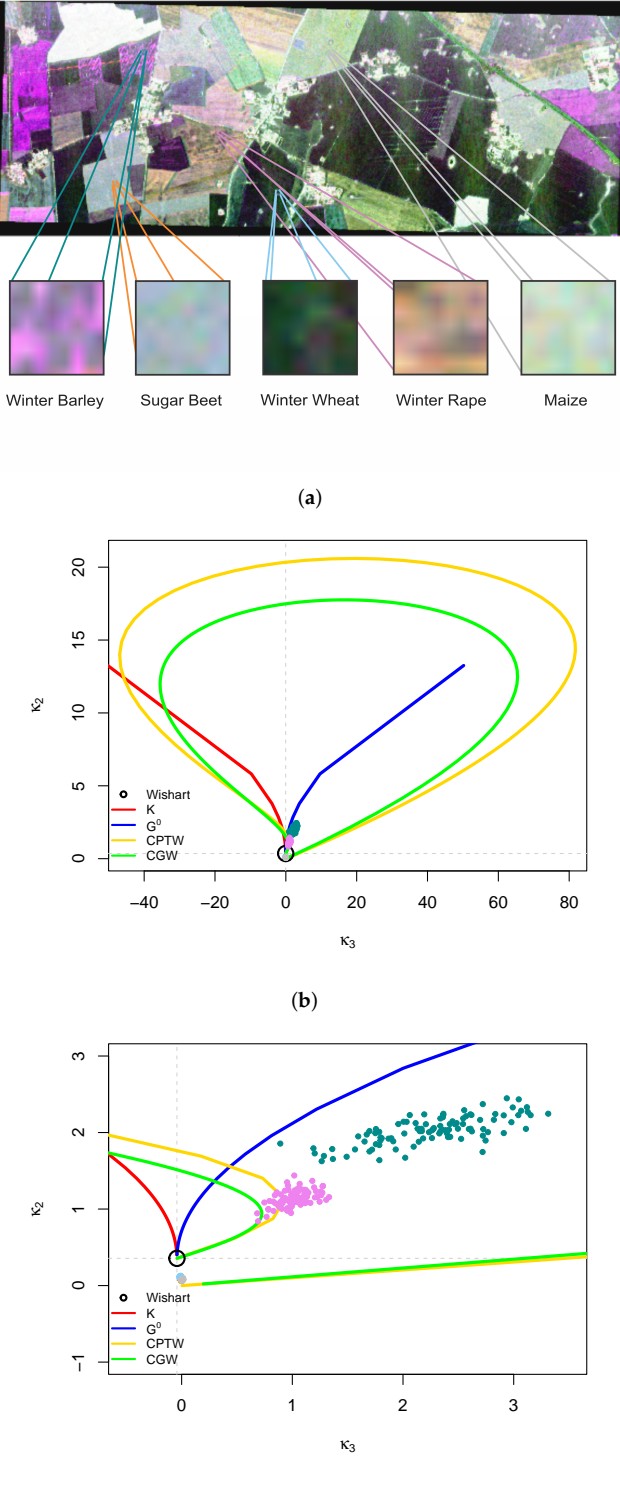

**Figure 10.** MLC diagram with sample MLCs computed from the samples of AgriSAR. (**a**) Subset areas of AgriSAR. (**b**) MLC diagram with sample MLCs computed from the samples in (**a**). (**c**) Zoom on the data.

Finally, in order to complement the adequacy study of polarimetric distributions, we compared the fits of their marginal models. The marginal densities of the CTPC$\mathcal{W}_m^{\mathbb{C}}$ and CGC$\mathcal{W}_m^{\mathbb{C}}$ distributions are given in Equations (2) and (3), while those due to s$\mathcal{W}_m^{\mathbb{C}}$ and $\mathcal{K}_m$, $\mathcal{G}_m^0$ are the $\Gamma$ [24], $\mathcal{K}$ [37] and $\mathcal{G}^0$ [38] laws. These laws were employed to describe the intensities related to HH, HV and VV channels of the used images, considering ENL fixed.

We estimated the parameters of the proposed models, i.e., those characterised by densities (2) and (3), by way of the moments method. Let $z_1, \ldots, z_N$ be the observed intensity returns at a polarisation channel, and denote $\overline{Z} = N^{-1} \sum_{i=1}^{N} z_i$ and $\overline{Z^2} = N^{-1} \sum_{i=1}^{N} z_i^2$:

- The MoM estimates for $(\lambda, \sigma^2)$ in (2), $(\widehat{\lambda}, \widehat{\sigma^2})$, are given by:

$$\widehat{\sigma^2} = \frac{1}{2} \frac{\overline{Z} \left( e^{\widehat{\lambda}} - 1 \right)}{L \, \widehat{\lambda} \, e^{\widehat{\lambda}}}$$

and $\widehat{\lambda}$ is a solution of the nonlinear equation:

$$\overline{Z^2} = \frac{\overline{Z}^2 \left( e^{\widehat{\lambda}} - 1 \right)}{L \, \widehat{\lambda} \, e^{\widehat{\lambda}}} \left[ (L+1) + \widehat{\lambda} \, L \right].$$

- The MoM estimates for $(p, \sigma^2)$ in (3), $(\widehat{p}, \widehat{\sigma^2})$, are given by:

$$\widehat{p} = \frac{2 \, \widehat{\sigma^2}}{\overline{Z}} \quad \text{and} \quad \widehat{\sigma^2} = \frac{\overline{Z}^2 - \frac{1}{2} \overline{Z^2}}{\overline{Z}(L-1)},$$

subjected to the constraint $\overline{Z}^{-2} \overline{Z^2} < 2$ (condition that was verified for all used data).

Tables 5–7 show values of the Kolmogorov–Smirnov statistic (and its associated *p*-value), $S_{\mathrm{KS}}$, and corrected Akaike information criterion ($\mathrm{AIC}_c$) for San Francisco, Foulum and Neubrandenburg images. It is known that the first comparison measure assesses the fit to the empirical cumulative distribution function, while the second defines a comparison criterion in terms of empirical densities.

**Table 5.** Results of fitting intensities of San Francisco PolSAR data, for $L$ fixed.

| Region | AIC$_c$ Values | | | | | KS Statistics (*p*-Value) | | | | |
|---|---|---|---|---|---|---|---|---|---|---|
| | $\Gamma$ | $\mathcal{K}$ | $\mathcal{G}^0$ | CTPCW | CGCW | $\Gamma$ | $\mathcal{K}$ | $\mathcal{G}^0$ | CTPCW | CGCW |
| Ocean HH | −3336.182 | −3353.678 | −3350.913 | −3349.626 | −3349.205 | 0.0463 (0.3352) | 0.0436 (0.4064) | 0.0339 (0.7228) | 0.0354 (0.6736) | 0.0339 (0.7257) |
| Ocean HV | −5370.004 | −5367.173 | −5370.251 | −5371.358 | −5371.422 | 0.0341 (0.7195) | 0.0361 (0.6481) | 0.0233 (0.9774) | 0.0213 (0.9914) | 0.0211 (0.9924) |
| Ocean VV | −2449.256 | −2460.579 | −2459.775 | −2457.083 | −2456.303 | 0.0623 (0.0788) | 0.0447 (0.3767) | 0.0468 (0.3201) | 0.0499 (0.2506) | 0.0505 (0.2385) |
| Forest HH | −778.383 | −1678.613 | −1756.631 | −1123.571 | −1644.802 | 0.2953 (0.0000) | 0.0842 (0.0004) | 0.02656 (0.7909) | 0.0924 (0.0000) | 0.0791 (0.0011) |
| Forest HV | −1920.739 | −2657.491 | −2704.299 | −1987.513 | −2616.529 | 0.2420 (0.0000) | 0.0424 (0.2303) | 0.0318 (0.5763) | 0.0769 (0.0016) | 0.0221 (0.9319) |
| Forest VV | −780.561 | −1702.588 | −1757.682 | −1135.981 | −1674.072 | 0.2862 (0.0000) | 0.0717 (0.0041) | 0.0241 (0.8753) | 0.0878 (0.0002) | 0.0644 (0.0137) |
| Urban HH | 1109.161 | −61.096 | −150.191 | −613.3867 | −604.9043 | 0.4206 (0.0000) | 0.0996 (0.0013) | 0.0294 (0.9063) | 0.0622 (0.2362) | 0.0702 (0.1313) |
| Urban HV | −396.151 | −1215.056 | −1257.915 | −1371.682 | −1369.557 | 0.3222 (0.0000) | 0.0728 (0.0405) | 0.0371 (0.6894) | 0.0898 (0.0232) | 0.0918 (0.0189) |
| Urban VV | 549.423 | −178.221 | −191.749 | −587.054 | −563.3056 | 0.3283 (0.000) | 0.0641 (0.0976) | 0.0555 (0.2057) | 0.0572 (0.3256) | 0.0767 (0.0775) |

From Table 5, the $\mathcal{K}$ and CGCW marginals show the best results for ocean regions. The $\mathcal{G}^0$ distribution yields the best fits for forest regions; with the exception of the HV channel which is best represented by the CGCW distribution. The best characterisations for urban scenarios are made by the $\mathcal{G}^0$ and CPTCW distributions.

From Table 6, the $\mathcal{G}^0$, CPTCW and CGCW marginals obtain the best results for Background, Rape and Wheat regions. The Oat region is best characterised by the proposed models; with exception of the VV channel, on which the $\Gamma$ law presents the smallest $\mathrm{AIC}_c$. Our models suggest the best fits for Rye and Conifer regions.

From Table 7, the $\Gamma$ marginal shows the best results for the beet region. The $\Gamma$ and CGCW obtain the best performance for the Maize region. The $\mathcal{G}^0$ and proposed models achieve the best fits for the Rape and Barley regions. The $\mathcal{K}$, $\mathcal{G}^0$ and CGCW laws provide the best descriptions of Wheat.

Figure 11 shows three empirical and fitted density plots for the San Francisco urban, Foulum background and DEMMIN-Gormin winter rape images. We computed the histograms using the Freedman and Diaconis [39] rule. It is noticeable that the proposed marginal distributions tend to produce more flexible curves than the classical ones. In San Francisco urban scenarios, the CTPCW and $\mathcal{G}_I^0$ distributions furnish the best adher-

ence by the KS statistics. In the Foulum background areas, the CTPCW and CGCW laws perform better than the others. The best fits to DEMMIN-Gormin rape scenes are produced by the $\mathcal{G}^0$ and CTPCW laws. Additional fits for other scenarios can be found in https://www.dropbox.com/s/bjpcri9jfgtxglp/Graphs.pdf?dl=0 (accessed on 3 October 2022).

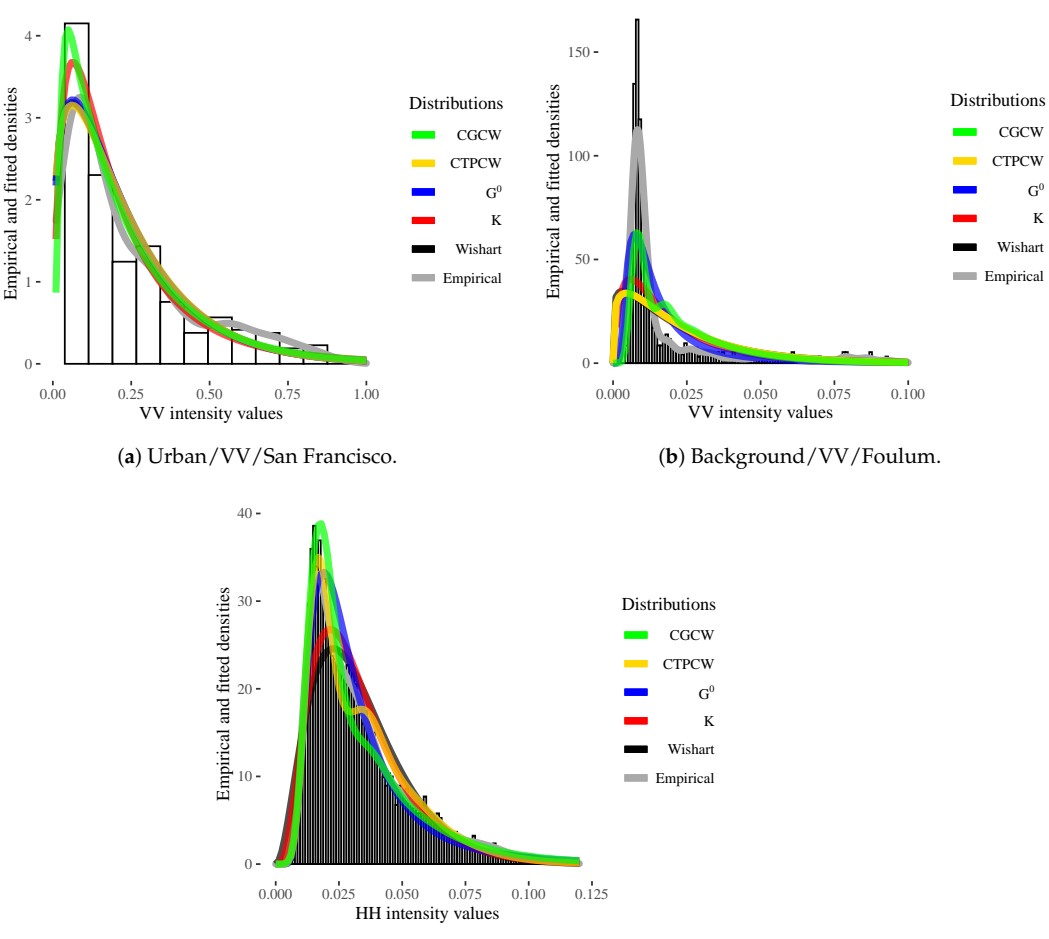

(**a**) Urban/VV/San Francisco.

(**b**) Background/VV/Foulum.

(**c**) Winter Rape/HH/DEMMIN-Gormin.

**Figure 11.** Empirical and fitted densities for San Francisco, Foulum and DEMMIN-Gormin images.

**Table 6.** Results of fitting intensities of Foulum PolSAR data, for L fixed.

| Region | AIC$_c$ Values | | | | | KS Statistics ($p$-Value) | | | | |
|---|---|---|---|---|---|---|---|---|---|---|
| | $\Gamma$ | $\mathcal{K}$ | $\mathcal{G}^0$ | CPTCW | CGCW | $\Gamma$ | $\mathcal{K}$ | $\mathcal{G}^0$ | CTPCW | CGCW |
| Back. HH | 13,644.297 | −5662.811 | −6727.839 | 37,551.191 | 31,703.135 | 0.7328 (0.0000) | 0.3375 (0.0000) | 0.2438 (0.0000) | 0.2567 (0.0000) | 0.2536 (0.0000) |
| Back. HV | 15,795.002 | −8111.433 | −9317.474 | 31,848.566 | 19,651.018 | 0.7509 (0.0000) | 0.3682 (0.0000) | 0.2684 (0.0000) | 0.2541 (0.0000) | 0.2701 (0.0000) |
| Back. VV | −1086.881 | −6128.267 | −6639.634 | −3668.218 | −6014.533 | 0.5713 (0) | 0.2425 (0.0000) | 0.1813 (0.0000) | 0.1758 (0.0000) | 0.1557 (0.0000) |
| Rape HH | −15,538.403 | −16,372.233 | −16,398.015 | −16,324.445 | −16,311.755 | 0.1189 (0.0000) | 0.0178 (0.2733) | 0.0134 (0.6230) | 0.0362 (0.0005) | 0.0391 (0.0001) |
| Rape HV | −28,160.355 | −28,834.923 | −28,383.304 | −28,872.724 | −28,849.118 | 0.1202 (0.0000) | 0.0403 (0.0000) | 0.1311 (0.0000) | 0.0326 (0.0026) | 0.0365 (0.0000) |
| Rape VV | −15,421.004 | −15,404.677 | −15,462.128 | −15,490.977 | −15,493.038 | 0.0369 (0.0004) | 0.0409 (0.0000) | 0.0292 (0.0099) | 0.0217 (0.1052) | 0.0210 (0.1276) |
| Wheat HH | −13,267.144 | −13,224.354 | −13,226.610 | −13,323.948 | −13,324.126 | 0.0299 (0.0222) | 0.0390 (0.0007) | 0.0475 (0.0000) | 0.0152 (0.6067) | 0.0153 (0.5949) |
| Wheat HV | −24,981.209 | −26,733.414 | −27,045.345 | −26,693.55 | −26,979.18 | 0.1790 (0.0000) | 0.1032 (0.0000) | 0.1254 (0.0000) | 0.0272 (0.0480) | 0.0291 (0.0279) |
| Wheat VV | −9490.612 | −9360.921 | −9420.177 | −9383.994 | −9490.997 | 0.0165 (0.4944) | 0.0577 (0.0000) | 0.0481 (0.0000) | 0.0358 (0.0031) | 0.0151 (0.6164) |
| Oat HH | −35,266.728 | −34,939.282 | −35,116.783 | −35,324.396 | −35,154.303 | 0.1344 (0.0000) | 0.1584 (0.0000) | 0.1441 (0.0000) | 0.1201 (0.0000) | 0.1621 (0.0000) |
| Oat HV | −43,082.474 | −42,739.723 | −42,570.853 | −42,910.284 | −43,067.083 | 0.1138 (0.0000) | 0.1493 (0.0000) | 0.1873 (0.0000) | 0.1048 (0.0000) | 0.1209 (0.0000) |
| Oat VV | −32,614.112 | −32,726.2166 | −32,807.781 | −33,303.94 | −33,335.41 | 0.1097 (0.0000) | 0.1089 (0.0000) | 0.0944 (0.0000) | 0.0599 (0.0000) | 0.0593 (0.0000) |
| Rye HH | −28,348.174 | −28,174.571 | −27,917.223 | −28,602.34 | −28,604.95 | 0.0871 (0) | 0.1158 (0) | 0.1631 (0) | 0.0647 (0.0000) | 0.0648 (0.0000) |
| Rye HV | −32,959.766 | −32,823.334 | −32,481.953 | −33,148.64 | −33,149.46 | 0.0717 (0.0000) | 0.0961 (0.0000) | 0.1505 (0.0000) | 0.0469 (0.0000) | 0.0468 (0.0000) |
| Rye VV | −21,953.243 | −21,918.265 | −21,726.364 | −22,325.60 | −22,339.67 | 0.0946 (0.0000) | 0.0988 (0.0000) | 0.1544 (0.0000) | 0.0622 (0.0000) | 0.0597 (0.0000) |
| Conifer HH | −3104.633 | −2979.998 | −2998.099 | −3075.352 | −3143.367 | 0.0435 (0.0049) | 0.0805 (0.0000) | 0.0801 (0.0000) | 0.0338 (0.0535) | 0.0158 (0.8235) |
| Conifer HV | −6112.789 | −5910.137 | −5966.941 | −6121.835 | −6206.274 | 0.0731 (0.0000) | 0.1054 (0.0000) | 0.1080 (0.0000) | 0.0586 (0.0000) | 0.0585 (0.0000) |
| Conifer VV | −6290.114 | −6074.269 | −5997.672 | −6302.553 | −6394.536 | 0.0809 (0.0000) | 0.1133 (0.0000) | 0.1411 (0.0000) | 0.0692 (0.0000) | 0.0654 (0.0000) |

**Table 7.** Results of fitting intensities on intensities of AgriSAR PolSAR data for L fixed.

| Region | AIC$_c$ Values | | | | | KS Statistics ($p$-Value) | | | | |
|---|---|---|---|---|---|---|---|---|---|---|
| | $\Gamma$ | $\mathcal{K}$ | $\mathcal{G}^0$ | CTPCW | CGCW | $\Gamma$ | $\mathcal{K}$ | $\mathcal{G}^0$ | CTPCW | CGCW |
| Maize HH | −78,397.154 | −76,497.874 | −76,358.346 | −77,111.508 | −78,295.695 | 0.0878 (0.0000) | 0.1223 (0.0000) | 0.1431 (0.0000) | 0.1002 (0.0000) | 0.1032 (0.0000) |
| Maize HV | −94,634.188 | −93,234.909 | −93,275.192 | −95,332.257 | −95,387.211 | 0.0891 (0.0000) | 0.1264 (0.0000) | 0.1459 (0.0000) | 0.0972 (0.0000) | 0.0972 (0.0000) |
| Maize VV | −85,862.424 | −85,117.185 | −84,061.901 | −86,853.034 | −86,969.596 | 0.0688 (0.0000) | 0.1083 (0.0000) | 0.1523 (0.0000) | 0.0871 (0.0000) | 0.0865 (0.0000) |
| Rape HH | −47,307.555 | −60,318.864 | −60,769.119 | −61,066.01 | −60,758.62 | 0.2434 (0.0000) | 0.0579 (0.0000) | 0.0414 (0.0000) | 0.0170 (0.0029) | 0.0601 (0.0000) |
| Rape HV | −79,878.920 | −99,756.999 | −100,348.078 | −99,320.55 | −100,274 | 0.2661 (0.0000) | 0.0471 (0.0000) | 0.0216 (0.0000) | 0.0338 (0.0000) | 0.0256 (0.0000) |
| Rape VV | −79,273.791 | −82,175.302 | −81,683.624 | −82,827.037 | −82,867.0421 | 0.1199 (0.0000) | 0.0449 (0.0000) | 0.0884 (0.0000) | 0.0171 (0.0027) | 0.0222 (0.0000) |
| Wheat HH | −97,953.769 | −102,484.074 | −104,142.925 | −101,075.289 | −101,922.446 | 0.1055 (0.0000) | 0.0981 (0.0000) | 0.0763 (0.0000) | 0.0142 (0.0358) | 0.0142 (0.0359) |
| Wheat HV | −106,542.769 | −125,850.838 | −129,465.447 | −122,424.1 | −124,596.9 | 0.2177 (0.0000) | 0.1262 (0.0000) | 0.0938 (0.0000) | 0.0370 (0.0000) | 0.0367 (0.0000) |
| Wheat VV | −100,755.443 | −101,241.792 | −100,901.263 | −97,871.733 | −101,055.643 | 0.0732 (0.0000) | 0.0954 (0.0000) | 0.1438 (0.0000) | 0.0794 (0.0000) | 0.0681 (0.0000) |
| Beet HH | −78,449.372 | −76,431.501 | −75,912.814 | −77,602.605 | −78,078.731 | 0.1107 (0.0000) | 0.1455 (0.0000) | 0.1651 (0.0000) | 0.1253 (0.0000) | 0.1294 (0.0000) |
| Beet HV | −98,717.807 | −97,296.974 | −96,563.392 | −97,709.887 | −98,179.681 | 0.1638 (0.0000) | 0.1806 (0.0000) | 0.2059 (0.0000) | 0.1784 (0.0000) | 0.1733 (0.0000) |
| Beet VV | −77,030.751 | −74,324.474 | −74,858.236 | −76,188.637 | −76,452.111 | 0.1601 (0.0000) | 0.1941 (0.0000) | 0.1901 (0.0000) | 0.1672 (0.0000) | 0.1821 (0.0000) |
| Barley HH | 78,156.057 | −40,689.082 | −44,221.803 | −1,270.399 | −33,982.76 | 0.5524 (0.0000) | 0.14509 (0.0000) | 0.0518 (0.0000) | 0.1587 (0.0000) | 0.0771 (0.0000) |
| Barley HV | 59,700.677 | −98,734.402 | −111,375.215 | 12,029.802 | −26,949.616 | 0.7236 (0.0000) | 0.2588 (0.0000) | 0.1231 (0.0000) | 0.1031 (0.0000) | 0.1043 (0.0000) |
| Barley VV | 86,618.097 | −37,226.084 | −42,060.558 | 2,1926.804 | −22,934.483 | 0.5717 (0.0000) | 0.1575 (0.0000) | 0.0521 (0.0000) | 0.1339 (0.0000) | 0.1078 (0.0000) |

Finally, to illustrate the fitting in a bimodal scenario (beyond regions having only one texture), we select a region from a Foulum image obtained in VV polarisation. Figure 12a displays the selected part, while Figure 12b shows the histogram and the fitted (Wishart, $\mathcal{K}$, $\mathcal{G}^0$, CTPCW and CGCW) models. By visual inspection, one can observe that only our proposals recognise the bimodality present in the data, the CTPCW distribution in particular is the closest to the empirical density.

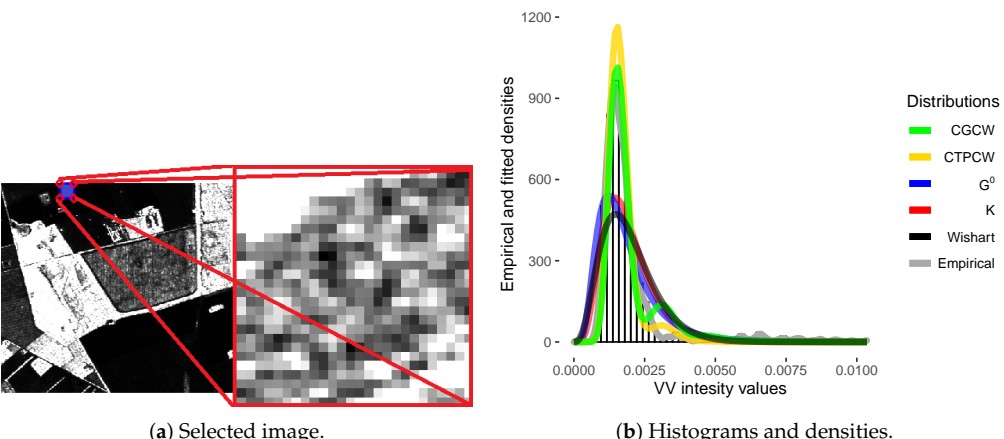

(**a**) Selected image.      (**b**) Histograms and densities.

**Figure 12.** Selected image and empirical and fitted histograms.

## 5. Conclusions

In this paper, we have proposed two new PolSAR distributions called compound truncated Poisson complex Wishart (CTPC$\mathcal{W}_m^{\mathbb{C}}$) and compound Geometric complex Wishart (CGC$\mathcal{W}_m^{\mathbb{C}}$). They have been connected with the physical formation of the PolSAR system by way of the compound matrix summation method. Some of their properties have been derived: characteristic function and Mellin-kind log-cumulant (MLC). To fit our PolSAR models in practice, we have proposed maximum likelihood estimators (MLEs) equipped by the expectation maximisation algorithm. The closed-form expressions for MLEs have been found (avoiding errors numerical fluctuations) and numerical results have indicated that such estimates presented low values of bias and mean square errors for sample sizes which are realistic with the PolSAR data processing practice. Three actual experiments with PolSAR images have been performed. Adopting the diagrams of MLCs as a comparison criterion, results pointed out CTPC$\mathcal{W}_m^{\mathbb{C}}$ and CGC$\mathcal{W}_m^{\mathbb{C}}$ models may provide better descriptions of some PolSAR scenarios than other well-known laws; e.g., s$\mathcal{W}_m^{\mathbb{C}}$, $\mathcal{K}_m$, and $\mathcal{G}_m^0$.

The proposed models open new venues of research, e.g., other estimators and their properties (including robust techniques), applied studies of a large size and heterogeneous samples, test statistics, their relationship with Information Theory and Information Geometry, noise reduction and clustering, among others.

The main drawback of the new models is that they require defining the weights of mixtures like a function of their components, thus limiting their flexibility. It is possible that other discrete variables for $N$ in (1) can solve this problem.

**Supplementary Materials:** Details of the proof of theoretical results are at https://www.dropbox.com/s/q5wotc6d5niqgv3/apendix.pdf?dl=0, accessed on 5 September 2022.

**Author Contributions:** All authors discussed the results and contributed to all sections. All authors have read and agreed to the published version of the manuscript.

**Funding:** This research was funded by CNPq (Conselho Nacional de Desenvolvimento Científico e Tecnológico), grant numbers 303267/2019-4 and 309538/2021-1.

**Institutional Review Board Statement:** Not applicable.

**Informed Consent Statement:** Not applicable.

**Data Availability Statement:** The data we used in this paper were obtained from https://earth.esa.int/eogateway accessed on 9 July 2020. Additional computational issues may be sent to email jodavid.arts@gmail.com.

**Acknowledgments:** The authors would like to thank Fundação do Amparo a Ciência e Tecnologia (FACEPE) and Conselho Nacional de Desenvolvimento Científico e Tecnológico (CNPq) for the support, and Centro Nacional de Supercomputação (CESUP), Universidade Federal do Rio Grande do Sul (UFRGS) for the support with the simulations and applications.

**Conflicts of Interest:** The authors declare no conflict of interest.

## Abbreviations

The following abbreviations are used in this manuscript:

| | |
|---|---|
| PolSAR | Polarimetric synthetic aperture radar |
| CTPCW | compound truncated Poisson complex Wishart |
| CGCW | d compound geometric complex Wishart |
| MLCs | Mellin-kind log-cumulants |
| SCM | sample covariance matrices |
| MM | multiplicative modeling |
| cf | characteristic function |
| MLEs | maximum likelihood estimators |

| EM | Expectation Maximisation |
|---|---|
| MoM | Moment Method |
| pmf | probability mass function |
| pdf | probability density function |
| MKS | Mellin-kind statistic |
| MCGF | Mellin-kind cumulant-generating |
| AIRSAR | Airborne Synthetic Aperture Radar |
| EMISAR | SAR image system of the Electromagnetics Institute |

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
