# Peer review of "PolSAR Models with Multimodal Intensities"

_remotesensing, doi:10.3390/rs14205083_

Round 1

Reviewer 1 Report

In this investigation, the authors propose models that comply with the physics of the formation of PolSAR images. These models are useful to express multimodal data for SAR images, which are often contaminated with multiplicative speckle noise. The authors employ the proposed models to describe PolSAR images and demonstrate their ability to outperform the existing distribution models. 

Introduction
The authors are requested to discuss the work of researchers like J.M. Nicolas, F. Tupin, V. Krylov, and G. Moser, who have extensively worked on the topic of statistical modelling of SAR images including multimodal distributions and MoLC. Furthermore, the authors are requested to indicate the gap/shortcomings in their work that leaves room for the contributions in the present investigation.

Methodology and Results
i) The authors are requested to demonstrate how the proposed models prove that the assumptions of additivity of noise and Gaussianity of distributions are indeed incorrect while handling SAR images. Can you suggest a simple polarimetric indicator to test these assumptions?

ii) The authors have mentioned multimodal distributions as one of the key strengths of the proposed models. The reviewer would like to request the authors to present such multimodal fitting with real SAR data. Fig. 11 only indicates largely a single modal fitting and the multimodal fitting is not very prominent (only visible in Wishart). The authors are requested to choose an appropriate region of interest where the histogram will indicate a bi-modal distribution.

Some other comments are as follows:
i) Provide Pauli RGB of the AIRSAR image and other sites/sensors right
at the beginning.

ii) Provide enlarged versions of Fig. 5, 6, and 7. The colour bar range
is difficult to read.

iii) Consider reducing the mathematical equations and supplementing them
with references.

iv) It is a good idea to bunch all additional material (like pdf links)
at the end of the paper and call it Supplementary.

v) If possible consider presenting the results of the fitting
intensities in a graphical form.

Reviewer 2 Report

Good work. I only have one suggestion. Experiments need to be performed on data from typical satellite sensors with a relatively large scene. The data used in this manuscript only covers a very small area, which is not good to oberserve the performance of the methods. 

Reviewer 3 Report

The manuscript introduces two new Pol-SAR distributions and the experiments over three real polarimetric SAR datasets, acquired using AIRSAR, E-SAR, and EMISAR sensors, prove the efficacy of the proposed models. I recommend the paper for publication subject to the clarification of the following points:

(1)  I appreciate that the well-known ocean, urban, and forest regions of the AIRSAR San Francisco dataset are taken under consideration for the analysis. However, for one set of result evaluations, the three regions named "A1", "A2", and "A3" in the manuscript, representing the ocean, forest, and urban, respectively, are taken under consideration (as shown in Fig. 4), whereas, for another parametric result evaluation, again ocean, urban, and forest regions (named as, water, vegetation, and urban, respectively, in Fig. 8(a)) are selected, but in this case, the chosen regions are completely different than the previous three regions. To maintain consistency, I request the authors to either select the same set of regions or explain why this difference is important in the manuscript.

(2)  In the selected Foulum region dataset of EMISAR, the areas containing different crops-type are taken under consideration, as clearly mentioned in Fig. 9(a). I suggest you to add the reference or URL in the manuscript that validates the ground truth about the crops mentioned in Fig. 9(a). This will be a great help to the reader of the manuscript to directly relate to the results. 

Minor Corrections:

Line 38: " . . this unique feature had only be previously obtained by the use of mixture models. . . " The "be" should be replaced with "been".

Line 180: ". . .Figs. 5–7 show maps of thse estimates for San Francisco . . ." Spelling mistake in this line needs to be corrected. The "thse" should be replaced with "these".
